# Revisiting Pre-Propagation GNNs: Robust Diffusion Operators and Hidden-State Re-Propagation

**Zichao Yue** [1]   **Zhiru Zhang** [1]

## Abstract

Pre-propagation graph neural networks (PP-GNNs) decouple node feature propagation from transformation: graph diffusion is performed once as preprocessing, and training reduces to dense per-node transformations. This design enables mini-batch training without inter-node dependencies, avoids repeated sparse matrix–matrix multiplications, and better matches modern accelerators optimized for dense compute. However, their expressivity remains unclear, and empirical results show a gap between PP-GNNs and their message-passing counterparts on commonly used graph benchmarks, especially heterophilic ones. In this paper, we propose a suite of robust graph diffusion operators for preprocessing and a few-shot hidden-state re-propagation scheme during training. Our methods improve the validation and test accuracy of PP-GNNs, enabling them to match the accuracy of message-passing GNNs while maintaining training efficiency.

## 1. Introduction

The message-passing scheme (Gilmer et al., 2017) underlies most graph neural networks (GNNs), but its memory and computation costs grow exponentially with neighborhood expansion (Hamilton et al., 2017), making large-scale training challenging. Prepropagation GNNs (PP-GNNs) have emerged as a scalable alternative (Wu et al., 2019; Frasca et al., 2020; Dong et al., 2021; Zhang et al., 2022; Liao et al., 2022; Chen et al., 2020b; Deng et al., 2024; Zhu & Koniusz, 2021) by decoupling feature propagation from transformation. In PP-GNNs, node features are diffused over the graph once as a preprocessing step, and learning proceeds on the resulting diffused features via dense per-node transforma-

*Table 1.* Motivating accuracy gap between MP-GNNs and PP-GNNs on heterophilic benchmarks. Entries report mean test accuracy on `roman-empire` and ROC AUC on `minesweeper`.

| Model | Minesweeper | Roman-Empire |
|---|---|---|
| **MP-GNNs** | | |
| SAGE | $93.51 \pm 0.57$ | $85.74 \pm 0.67$ |
| GAT | $93.91 \pm 0.35$ | $88.75 \pm 0.41$ |
| DIR-GNN | $87.05 \pm 0.69$ | $91.23 \pm 0.32$ |
| **PP-GNNs** | | |
| SIGN | $90.71 \pm 0.56$ | $80.01 \pm 0.50$ |
| HOGA | $90.53 \pm 0.66$ | $79.39 \pm 0.56$ |
| GAMLP | $90.47 \pm 0.66$ | $78.87 \pm 0.65$ |
| **Best MP – Best PP gap** | **3.20** | **11.22** |

tions. This decoupling eliminates inter-node dependencies during training, making mini-batch optimization straightforward, avoiding repeated sparse matrix–matrix multiplications (SpMM), and better matching modern accelerators optimized for dense compute, while substantially improving efficiency and scalability compared to message-passing GNNs (MP-GNNs) (Yue et al., 2025).

PP-GNNs show competitive performance on large graph benchmarks (e.g., OGB), which are predominantly homophilic graphs. However, recent theoretical study suggests the scalability advantage of PP-GNNs may come with a sacrifice in expressivity (Chen et al., 2020a). Empirically, we find this limitation is most pronounced on heterophilic graphs: as shown in Table 1, the test accuracy gap between MP-GNNs and PP-GNNs can reach up to 11% on several heterophilic benchmarks. We hypothesize that two design aspects of PP-GNNs contribute to this gap: (i) diffusion operator limitations and (ii) one-shot feature propagation, and address them with more robust diffusion operators and hidden-state re-propagation (HRP).

**First, diffusion-operator limitations.** A central design choice in PP-GNNs is the *preprocessing diffusion operator* that generates multi-hop features. Most existing PP-GNNs adopt simple operators—e.g., normalized adjacency or random-walk variants—whose spectral responses behave largely as *low-pass* graph filters (Nt & Maehara, 2019; Gasteiger et al., 2019b), which can be suboptimal on heterophilic graphs that benefit from higher-frequency or band-pass components (Chien et al., 2021; Bo et al., 2021). In principle, combining hop features from multi-step diffu-

[1] School of Electrical and Computer Engineering, Cornell University, Ithaca, New York, USA. Correspondence to: Zichao Yue <zy383@cornell.edu>.

*Proceedings of the $43^{rd}$ International Conference on Machine Learning*, Seoul, South Korea. PMLR 306, 2026. Copyright 2026 by the author(s).

sion (e.g., $\{\Phi^k X\}_{k=0}^K$) enables richer spectral shaping; in practice, the hop budget $K$ is typically kept small due to efficiency considerations and the *oversmoothing* issue (Li et al., 2018). Under such small $K$, the standard *monomial* diffusion basis induced by powers of $\Phi$ can be poorly conditioned and yield highly correlated hop features, limiting approximation quality and destabilizing the learned hop combination. To address this, we construct better-conditioned diffusion bases using *orthogonal-polynomial* recurrences and *Lanczos/Krylov* subspace methods (Jacobi-polynomial and Krylov-subspace bases), which reduce approximation error under the same hop budget and improve the stability of hop aggregation (Liao et al., 2019; Wang & Zhang, 2022).

**Second, one-shot propagation**. Although PP-GNNs incorporate multi-hop information in preprocessing, this propagation is performed only once on the raw inputs and is not coupled to evolving hidden representations. Inspired by polynomial-filter GNNs like ChebNet(Defferrard et al., 2016), where depth still increases expressivity even with large receptive fields per layer, we introduce hidden-state re-propagation to PP-GNNs, effectively adding additional rounds of graph filtering on task-adapted hidden states. A related idea is to reuse model predictions by treating logits—the learned soft labels—as additional node features (Wang et al., 2021; Zhang et al., 2022). However, we empirically show that replacing hidden states with logits in the re-propagation step leads to a substantial accuracy drop, suggesting that re-propagated hidden representations carry complementary information beyond class probabilities. In practice, only a few hidden-state re-propagation rounds are needed for convergence, preserving the training efficiency and scalability of vanilla PP-GNNs.

To summarize, we study how to close the accuracy gap of PP-GNNs and MP-GNNs, especially on heterophilic graphs, while preserving their training efficiency and scalability. Our approach improves PP-GNNs by (i) using more robust diffusion operators and (ii) lightweight recoupling of feature propagation with representation learning. While motivated by heterophily, we observe that these improvements also transfer to homophilic benchmarks, yielding gains on a majority of them. Our main contributions are as follows:

- We adapt Jacobi-polynomial and Krylov-subspace diffusion bases to the preprocessing regime of PP-GNNs, improving the conditioning of precomputed hop features and enabling richer spectral responses under a fixed hop budget.

- We introduce hidden-state re-propagation, a mechanism that recouples feature propagation with learned representations and outperforms label-based reuse.

- Across 4 PP-GNNs and 12 datasets, our method boosts test accuracy by $+\mathbf{2.18}$ avg on heterophilic ones and

$+\mathbf{1.96}$ avg on homophilic ones, outperforming MP-GNN baselines on **7** out of 12 datasets.

- Additionally, we explore an RNN-based hop aggregator as an efficient alternative to multi-head-attention-based aggregation, achieving comparable accuracy while significantly reducing end-to-end training time.

## 2. Preliminaries

### 2.1. Graph Notation

Let $G = (V, E)$ be a graph with $|V| = n$ nodes and adjacency matrix $A \in \mathbb{R}^{n \times n}$. Let $X \in \mathbb{R}^{n \times d}$ denote input node features and $Y \in \{1, \ldots, C\}^n$ node labels for a $C$-class node classification task. We use $\mathcal{V}_{\mathrm{tr}}, \mathcal{V}_{\mathrm{va}}, \mathcal{V}_{\mathrm{te}}$ to denote train/val/test node splits. Let $D = \mathrm{diag}(d_1, \ldots, d_n)$ be the degree matrix with $d_i = \sum_j A_{ij}$. For a node $i$, $\mathcal{N}(i)$ denotes its neighbor set. We write $\sigma(\cdot)$ for a pointwise nonlinearity and $\| \cdot \|$ for concatenation.

### 2.2. Graph Spectral Theory and Spectral Graph Filters

For an undirected graph, the normalized Laplacian is

$$L = I - D^{-1/2} A D^{-1/2}. \tag{1}$$

Since $L$ is real symmetric, it admits an eigendecomposition

$$L = U \Lambda U^\top, \tag{2}$$

where $U = [u_1, \ldots, u_n]$ is orthonormal and $\Lambda = \mathrm{diag}(\lambda_1, \ldots, \lambda_n)$ with $0 = \lambda_1 \leq \cdots \leq \lambda_n \leq 2$. Given a graph signal $x \in \mathbb{R}^n$ (e.g., one feature channel of $X$), its graph Fourier transform is $\hat{x} = U^\top x$ and inverse is $x = U\hat{x}$.

A *spectral graph filter* is defined by a scalar response function $g : \mathbb{R} \to \mathbb{R}$ applied to the Laplacian spectrum:

$$g(L)x = U\, g(\Lambda)\, U^\top x, \quad g(\Lambda) = \mathrm{diag}\big(g(\lambda_1), \ldots, g(\lambda_n)\big). \tag{3}$$

Intuitively, smaller eigenvalues correspond to smoother (low-frequency) components, while larger eigenvalues correspond to more oscillatory (high-frequency) components; $g(\lambda)$ specifies how each frequency band is attenuated.

In practice, explicit eigendecomposition can be avoided by approximating the matrix function $g(L)$ directly as an operator on $L$, rather than evaluating $g$ on individual eigenvalues. A common choice is a **polynomial filter**, which approximates $g(L)$ by a degree-$K$ polynomial in $L$:

$$g(L) \approx \sum_{k=0}^K \alpha_k\, p_k(L), \tag{4}$$

### 2.3. From Spectral Filters to Message-Passing GNNs

Spectral GNNs can be viewed as learning the filter $g(L)$ in (3). ChebNet-style models approximate $g(L)$ with truncated Chebyshev expansions as in (4), where $p_k(\cdot)$ is the

Chebyshev polynomial of degree $k$ and the coefficients $\{\alpha_k\}$ are learned.

GCN (Kipf & Welling, 2017) can be derived as a particular low-order ($K=1$) approximation of ChebNet, resulting in a degree-1 polynomial filter in the Laplacian $L$. Consequently, each GCN layer performs essentially one-hop neighborhood mixing; larger multi-hop receptive fields are obtained by stacking layers, yielding a propagation rule based on a normalized adjacency-like operator.

Abstractly, many GNNs implement repeated local aggregation and transformation, which can be written in the message-passing (MP) form

$$h_i^{(\ell+1)} = \phi^{(\ell)}\Big(h_i^{(\ell)}, \ \text{AGG}\big(\{\psi^{(\ell)}(h_i^{(\ell)}, h_j^{(\ell)}, e_{ij})\}\big)\Big). \quad (5)$$

where $j \in \mathcal{N}(i)$, $h_i^{(\ell)}$ is the hidden state at layer $\ell$, $e_{ij}$ denotes optional edge features, and AGG is a permutation-invariant aggregator. From this perspective, MP-GNNs can be interpreted as applying a graph filter to hidden representations interleaved with nonlinear transformations, bridging the spectral view and the spatial view.

### 2.4. Pre-Propagation GNNs and Graph Diffusion Operators

Pre-propagation GNNs (PP-GNNs) decouple propagation from learnable transformation. They first construct a *graph diffusion operator* $\Phi \in \mathbb{R}^{n \times n}$ from topology, and apply it to node features before training a dense predictor. Typical choices of $\Phi$ include normalized adjacency and random-walk operators.

Importantly, diffusion operators and spectral filters are closely related: when $\Phi$ can be written as a function of the Laplacian, i.e.,

$$\Phi = g(L), \quad (6)$$

then applying diffusion $\Phi X$ is exactly applying a spectral graph filter with response $g(\lambda)$ as in (3).

Given $\Phi$, PP-GNNs precompute propagated features via multi-step diffusion operations. Multi-hop features are then obtained as

$$Z = \big(X, \ \Phi X, \ \Phi^2 X, \ \ldots, \ \Phi^K X\big) \in \mathbb{R}^{n \times (K+1)d}, \quad (7)$$

which is then fed to a dense model $f_\theta$ (e.g., an MLP) to produce predictions:

$$H = f_\theta(Z), \qquad \hat{Y} = \text{softmax}(H). \quad (8)$$

PP-GNNs can be viewed as learning an implicit graph filter parameterized by $f_\theta$ over the subspace spanned by the diffusion powers of $\Phi$. Since $f_\theta$ does not perform message passing, training can be implemented with dense kernels and mini-batching over nodes, often improving scalability compared to MP-GNNs.

### 2.5. Label Usage as Input Features

Some methods augment node features with training label information. Specifically, we construct a label-feature matrix $L_{\text{in}} \in \mathbb{R}^{N \times C}$ whose $i$-th row is the one-hot encoding of $Y_i$ if node $i \in \mathcal{V}_{\text{tr}}$, and the zero vector otherwise. These label features (or *soft labels* such as logits/probabilities from a previous stage) can be concatenated with node features and optionally diffused using the same operator, e.g., $\Phi L_{\text{in}}$.

## 3. Method

**Overview.** We aim to improve the accuracy of PP-GNNs while preserving their scalability benefits. Our approach targets two complementary aspects of standard PP-GNNs: (i) the *diffusion operator* used to precompute multi-hop features, and (ii) the *one-shot* nature of pre-propagation that decouples propagation from evolving representations. Accordingly, we introduce (1) adoption of stronger graph diffusion operators with more robust diffusion bases, (2) a lightweight *hidden-state re-propagation* mechanism that recouples propagation with learned representations, and (3) an optional RNN-based hop aggregator with higher efficiency. The overview of method (1) and (2) are provided in Figure 1.

### 3.1. Robust diffusion operators

**Motivation.** Most PP-GNNs rely on simple diffusion operators, such as normalized adjacency or random-walk propagation, which typically exhibit low-pass smoothing behavior (Nt & Maehara, 2019; Gasteiger et al., 2019b). However, heterophily tasks can benefit from preserving or emphasizing higher-frequency and band-pass components (Chien et al., 2021; Bo et al., 2021). In principle, combining multi-hop features from a diffusion basis $\{\Phi^k X\}_{k=0}^{K}$ allows PP-GNNs to approximate richer spectral responses. In practice, however, the standard monomial hop bank is often highly correlated and poorly conditioned (Wang & Zhang, 2022); achieving accurate and stable approximations may therefore require larger hop budgets, which are limited by efficiency and *oversmoothing* considerations (Li et al., 2018).

These observations motivate a preprocessing-specific operator-design problem: how to construct a fixed diffusion bank that is well conditioned, spectrally diverse, and reusable throughout dense PP-GNN training. We address this problem by adapting Jacobi-polynomial and Krylov-subspace diffusion bases to the preprocessing regime of PP-GNNs. For Jacobi diffusion, we use an orthogonal polynomial basis on $[-1, 1]$ and calibrate its spectral emphasis once from graph-level statistics, reducing the need for validation-driven operator search during preprocessing. For Krylov diffusion, we construct feature-channel-adaptive Lanczos bases, which provide orthogonal, signal-aligned diffusion components and remain practical because they are computed

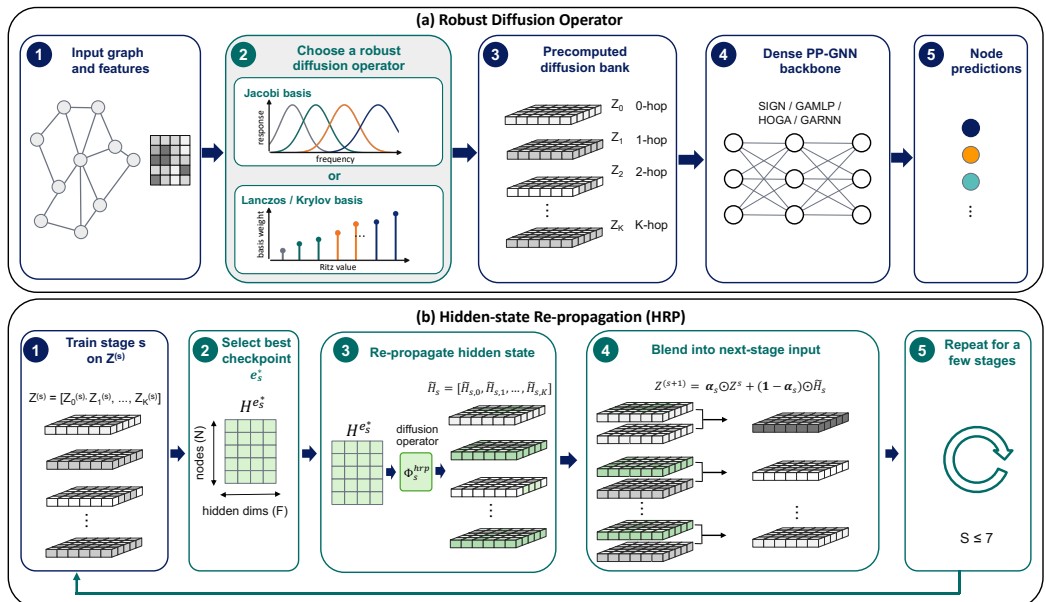

*Figure 1.* Overview of the proposed robust PP-GNN framework. (a) Robust diffusion operator: the key preprocessing step is to replace the standard monomial hop bank with a better-conditioned diffusion basis, using either a calibrated Jacobi basis or a channel-adaptive Lanczos/Krylov basis. This produces a precomputed hop-wise bank $Z = [Z_0, Z_1, \ldots, Z_K]$ that is fed to a dense PP-GNN backbone for node prediction. (b) Hidden-state re-propagation (HRP): at each stage, HRP selects the best hidden representation, re-propagates it to generate a hop-wise hidden-state bank, and blends it with the previous-stage input to form the next-stage bank.

once before training rather than recomputed across epochs. Together, these PP-GNN-oriented diffusion bases improve the conditioning and diversity of precomputed hop features, enabling richer spectral responses under a fixed hop budget while preserving the decoupled training pipeline.

### 3.1.1. JACOBI OPERATOR.

This diffusion operator constructs hop features using Jacobi polynomials, which form an orthogonal basis on $[-1, 1]$ under a weight determined by $(\alpha, \beta)$. Let $\{P_k^{(\alpha, \beta)}(\cdot)\}_{k \geq 0}$ denote Jacobi polynomials. We define the $k$-th filtered feature as

$$X_k = P_k^{(\alpha, \beta)}(\tilde{L}) X, \qquad k = 0, \ldots, K. \quad (9)$$

where $\tilde{L}$ is the shifted Laplacian $L - I$ that has spectrum in $[-1, 1]$, matching the canonical domain of classical orthogonal polynomials. Rather than explicitly forming $P_k(\tilde{L})$, we compute $X_k$ via the three-term recurrence

$$
\begin{aligned}
X_0 &= X, \\
X_1 &= \left(a_0 \tilde{L} + b_0 I\right) X_0, \\
X_{k+1} &= \left(a_k \tilde{L} + b_k I\right) X_k - c_k X_{k-1}, \qquad k \geq 1, \quad (10)
\end{aligned}
$$

where the scalars $a_k, b_k, c_k$ are determined by the Jacobi parameters $(\alpha, \beta)$ (standard closed-form coefficients). This yields a polynomial diffusion basis that can be better conditioned than the monomial basis $\{\tilde{L}^k X\}$, and shows better empirical performance (Wang & Zhang, 2022).

**Choice of Jacobi weights.** The parameters $(\alpha, \beta)$ control how the basis allocates "resolution" across the spectral interval $[-1, 1]$: informally, $\alpha$ emphasizes behavior near the high-frequency end ($\lambda \approx 1$) and $\beta$ near the low-frequency end ($\lambda \approx -1$). Two common special cases are:

$$
\begin{aligned}
(\alpha, \beta) &= (0, 0) \Rightarrow \text{Legendre}, \\
(\alpha, \beta) &= \left(-\tfrac{1}{2}, -\tfrac{1}{2}\right) \Rightarrow \text{Chebyshev}.
\end{aligned}
\quad (11)
$$

Beyond fixed choices, we adapt $(\alpha, \beta)$ to the input graph via a one-time *spectral calibration* step.

**One-time spectral calibration.** Although we cannot learn $(\alpha, \beta)$ during training due to the decoupling nature of PP-GNNs, we can still adapt them to a given graph by performing a lightweight, one-time calibration on $\tilde{L}$ during preprocessing. Using stochastic Chebyshev moments, we obtain a coarse estimate of the spectral density on $[-1, 1]$. We then compare the spectral mass in low- versus high-frequency regions and map their imbalance to Jacobi weights $(\alpha, \beta)$ via a simple monotone rule. Choosing $(\alpha, \beta)$ this way, rather than through validation-driven search, preserves the PP-GNN preprocessing contract by avoiding repeated data preparation and keeping the diffusion bank decoupled from training and reusable throughout dense optimization. Intuitively, the resulting basis allocates more resolution to the frequency region where the graph spectrum concentrates, while keeping calibration fully decoupled from training. Details are provided in Appendix E.

### 3.1.2. LANCZOS/KRYLOV OPERATOR.

While the Jacobi operator in (9) uses a *fixed* polynomial basis shared across all feature channels, we additionally consider a *data-adaptive* diffusion basis that (i) is orthogonal by construction and (ii) can capture channel-specific spectral content under a small hop budget. The key idea is to approximate the spectral action of $\tilde{L}$ on each channel via a low-rank Krylov subspace projection.

Let $X \in \mathbb{R}^{N \times D}$ be the node feature matrix and let $x_c \in \mathbb{R}^N$ denote its $c$-th channel. For an integer $m \ll N$, define the order-$m$ Krylov subspace with start vector $x_c$

$$\mathcal{K}_m(\tilde{L}, x_c) = \operatorname{span}\{x_c, \tilde{L}x_c, \ldots, \tilde{L}^{m-1}x_c\}. \quad (12)$$

Again, directly using the monomial vectors $\{\tilde{L}^k x_c\}$ is suboptimal (the vectors become increasingly collinear as $k$ grows). Instead, assuming $x_c \neq 0$, we run $m$ steps of the Lanczos iteration on the symmetric matrix $\tilde{L}$ with the normalized start vector $q_{c,1} = x_c/\|x_c\|$ and set $\beta_{c,0} = 0$ and $q_{c,0} = 0$. For $j = 1, 2, \ldots, m$, Lanczos constructs scalars $\alpha_{c,j} \in \mathbb{R}$ and $\beta_{c,j} \geq 0$ and the next basis vector $q_{c,j+1}$ via the three-term recurrence

$$\begin{aligned} r_{c,j} &= \tilde{L}q_{c,j} - \beta_{c,j-1}q_{c,j-1} - \alpha_{c,j}q_{c,j}, \\ \beta_{c,j} &= \|r_{c,j}\|, \qquad q_{c,j+1} = r_{c,j}/\beta_{c,j}, \end{aligned} \quad (13)$$

where $\alpha_{c,j} = q_{c,j}^\top \tilde{L}q_{c,j}$. Stacking $Q_c = [q_{c,1}, \ldots, q_{c,m}] \in \mathbb{R}^{N \times m}$ yields an orthonormal basis ($Q_c^\top Q_c = I_m$) and a symmetric tridiagonal matrix $T_c \in \mathbb{R}^{m \times m}$ with diagonal entries $\alpha_{c,j}$ and off-diagonal entries $\beta_{c,j}$, which represents the projection of $\tilde{L}$ onto $\mathcal{K}_m(\tilde{L}, x_c)$.

**Low-rank projection.** Given a spectral response $g(\cdot)$, we approximate the action of $g(\tilde{L})$ on $x_c$ by projecting $\tilde{L}$ onto the Krylov subspace and lifting the result back:

$$g(\tilde{L})\,x_c \approx Q_c\,g(T_c)\,Q_c^\top x_c, \qquad T_c = Q_c^\top \tilde{L}Q_c, \quad (14)$$

Consider the eigendecomposition of $T_c$:

$$T_c = U_c\Lambda_c U_c^\top, \qquad U_c = [u_{c,1}, \ldots, u_{c,m}] \in \mathbb{R}^{m \times m}, \quad (15)$$

where $\Lambda_c = \operatorname{diag}(\lambda_{c,1}, \ldots, \lambda_{c,m})$ contains the Ritz values. Then we choose $g$ to *select a single eigenmode* of $T_c$, which yields an explicit decomposition into decorrelated components. Specifically, for the $i$-th mode define a response that keeps only the Ritz value $\lambda_{c,i}$:

$$g_i(\Lambda_c) = \operatorname{diag}(0, \ldots, 0, 1, 0, \ldots, 0), \qquad g_i(T_c) = u_{c,i}u_{c,i}^\top \quad (16)$$

Combining with (14) gives the $i$-th *Ritz component* of $x_c$:

$$z_{c,i} := Q_c\,g_i(T_c)\,Q_c^\top x_c = Q_c(u_{c,i}u_{c,i}^\top)Q_c^\top x_c \quad (17)$$

Because the Lanczos start vector is $q_{c,1} = x_c/\|x_c\|$, we have $Q_c^\top x_c = \|x_c\|e_1$, and thus

$$z_{c,i} = \|x_c\|\,(u_{c,i}^\top e_1)\,Q_c u_{c,i} = (\|x_c\| \cdot u_{c,i,1})\,y_{c,i}, \quad (18)$$

where $u_{c,i,1}$ is the first entry of $u_{c,i}$, and $y_{c,i}$ is the *Ritz vectors* in the original node space.

**Interpretation.** The set $\{z_{c,i}\}_{i=1}^m$ forms a channel-wise *Ritz-component bank*: it decomposes $x_c$ into $m$ diffusion components within $\mathcal{K}_m(\tilde{L}, x_c)$, with mode shapes given by the lifted Ritz vectors $\{y_{c,i}\}$ and associated "frequencies" given by the Ritz values in $\Lambda_c$. Compared with the standard monomial hop bank $\{x_c, \tilde{L}x_c, \ldots, \tilde{L}^{m-1}x_c\}$, this component bank provides a better coordinate system for PP-GNN aggregation. Its orthogonality reduces correlations among hop features, its channel-specific start vector aligns the basis with the signal being filtered, and each response coefficient $g(\lambda_{c,i})$ directly weights one spectral component. These properties make the resulting diffusion bank easier for PP-GNN heads to aggregate under a fixed hop budget.

**Cost and practicality.** When Lanczos is run per channel, each iteration applies the same sparse operator $\tilde{L}$ to the current vectors; in practice, we batch all channels and compute $\tilde{L}Q_j$ using a single SpMM per step. Each step therefore costs $O(\operatorname{nnz}(\tilde{L})\,D)$, plus $O(ND)$ columnwise dot products and scalings to form $\alpha_{c,j}$, $\beta_{c,j}$, and normalize $q_{c,j+1}$. On sparse graphs with average degree $\bar{d} = \operatorname{nnz}/N$, the $O(ND)$ term is typically dominated by the SpMM, yielding an overall complexity $O(m\operatorname{nnz}(\tilde{L})\,D)$, the same leading order as $m$ steps of standard diffusion preprocessing. We also eigendecompose the small tridiagonal matrix $T_c \in \mathbb{R}^{m \times m}$ per channel, which costs $O(D\,m^3)$ and is negligible for $m \leq 15$ used in our settings. Forming all Ritz vectors via $Y_c = Q_c U_c$ costs $O(Nm^2)$ per channel, i.e., $O(Nm^2D)$ overall, comparable to the $m$ steps of diffusion cost.

### 3.2. Few-shot hidden-state re-propagation

**Motivation.** Even with stronger diffusion operators, the dense backbones of PP-GNNs can only learn on a fixed set of precomputed hop features. In contrast, MP-GNNs repeatedly interleave propagation with nonlinear transformations, so *task-adapted* representations are iteratively diffused and refined. To tackle this limitation, we introduce a lightweight *few-shot* re-propagation mechanism: we occasionally re-apply diffusion to intermediate hidden states during training, and feed the diffused representations back to the backbone, bridging one-shot PP-GNNs and iterative propagation while preserving dense-dominated training.

**Re-propagation mechanism.** Let $Z^{(s)}$ denote the hop-wise backbone input at stage $s$, and let $H^{(e)} \in \mathbb{R}^{N \times d}$ be the hidden states *before* the final output projection at epoch

$e$. We split training into $S$ stages; stage $s$ runs for $E_s$ epochs and is long enough for validation performance to stabilize. At the end of stage $s$, we select an epoch index $e_s^\star$ (default: the epoch attaining the best validation accuracy within the stage), detach the corresponding hidden representation $\bar{H}_s = \text{stopgrad}(H^{(e_s^\star)})$, and re-propagate it to form a hop-wise hidden-state bank:

$$\widetilde{H}_s = [\widetilde{H}_{s,0}, \widetilde{H}_{s,1}, \ldots, \widetilde{H}_{s,K}] = \mathcal{B}_K(\bar{H}_s; \Phi_s^{\text{hrp}}), \quad (19)$$

where $\Phi_s^{\text{hrp}}$ denotes the HRP diffusion operator and $\mathcal{B}_K(\cdot; \Phi_s^{\text{hrp}})$ denotes the corresponding $K$-hop diffusion-bank construction. For example, $\mathcal{B}_K$ can be instantiated either by a power-form bank $[\bar{H}_s, \Phi_s^{\text{hrp}}\bar{H}_s, \ldots, (\Phi_s^{\text{hrp}})^K\bar{H}_s]$ or by a Lanczos/Krylov-derived bank as in Section 3.1.2. We then augment the backbone input for the *next* stage by blending the previous-stage hop-wise input with the re-propagated hidden-state bank:

$$
\begin{aligned}
Z^{(s+1)} &= \boldsymbol{\alpha}_s \odot Z^{(s)} + \left(\mathbf{1} - \boldsymbol{\alpha}_s\right) \odot \widetilde{H}_s, \\
H^{(e)} &= f_\theta\left(Z^{(s)}\right), \qquad \text{for } e \in \text{stage } s.
\end{aligned}
\quad (20)
$$

with $Z^{(1)} = Z$. Here $\boldsymbol{\alpha}_s \in [0,1]^{K+1}$ is a hop-wise vector, and we set $\alpha_{s,0} = 1$ so that the 0-hop feature is kept unchanged, and blend every propagated hop $k \geq 1$ with its re-propagated hidden-state counterpart. The propagated-hop weights can be shared or hop-specific, and we optionally apply a cosine decay across stages so later stages place less weight on the re-propagated bank.

**Optional auto search.** We optionally enable a lightweight auto-search to reduce manual tuning of HRP. Specifically, we (i) tune the stage-wise blending weights with a simple evolution strategy, and (ii) select a stage checkpoint and re-propagation operator from a small candidate pool using a tight screening budget. We further promote diverse candidates by scoring how *spectrally different* a hidden state is from the original input via per-channel spectral moments of the normalized Laplacian. Additional details are provided in Appendix F.

**Efficiency.** Re-propagation occurs at low frequency ($S \leq 7$ in all experiments). Thus training remains dominated by dense computation in $f_\theta$, while diffusion is amortized across stages, preserving the scalability advantages of PP-GNNs.

### 3.3. RNN-based hop aggregator

**Motivation.** A critical design choice in PP-GNNs is how to aggregate hop-wise features. Existing backbones differ substantially: SIGN (Frasca et al., 2020) concatenates $\{Z_k\}_{k=0}^K$ and applies an MLP; GAMLP (Zhang et al., 2022) performs stage-wise attention over hops to emphasize informative diffusions; and HOGA (Deng et al., 2024) models

the hop axis as a sequence and applies multi-head attention (MHA). When $K$ is moderately large, MHA-like aggregators can incur non-trivial overhead. Motivated by the empirical observation that recurrent models can match MHA accuracy on short sequences at lower costs, we introduce an efficient RNN-based hop aggregator that treats hops as a sequence, following the sequential view in HOGA but with reduced compute and memory.

**Architecture.** Let $Z_k \in \mathbb{R}^{N \times d}$ denote the $k$-hop feature; we apply a shared recurrent update along the hop dimension:

$$S_k = \text{RNN}_\theta(Z_k, S_{k-1}), \qquad k = 0, \ldots, K, \quad (21)$$

where $S_k \in \mathbb{R}^{N \times h}$ is the recurrent state. We form the aggregated representation either from the last state or from a lightweight pooling over states:

$$Z_{\text{agg}} = S_K \quad \text{or} \quad Z_{\text{agg}} = \text{Pool}\left(\{S_k\}_{k=0}^K\right), \quad (22)$$

and feed $Z_{\text{agg}}$ to the dense predictor (MLP) for supervision. This hop aggregator is *optional* and can be paired with any diffusion family and with hidden-state re-propagation.

## 4. Experimental Setup

We conduct extensive experiments on both heterophilic and homophilic node classification benchmarks. We compare against a wide collection of MP-GNN baselines and show that, after incorporating our proposed diffusion operator suite and few-shot hidden-state re-propagation (HRP) across multiple training stages, PP-GNNs can match or surpass MP-GNNs while retaining the scalability benefits.

### 4.1. Datasets

We evaluate on six heterophilic graphs from (Lim et al., 2021; Platonov et al., 2023), which were curated to address common issues in earlier heterophilic benchmarks (e.g., unstable evaluation and split sensitivity). To verify that our approach is not restricted to heterophily, we additionally report results on six widely used homophilic datasets (Shchur et al., 1811; Hu et al., 2020). Dataset statistics and detailed split settings are provided in Appendix A.

### 4.2. Baselines

**Message-passing GNNs.** We compare against a broad set of standard message-passing GNN baselines (e.g., GCN/GAT/GraphSAGE and heterophily-oriented variants). For readability, we defer the mapping from baseline names to their original papers to Appendix C, as well as their hyperparameter settings.

**Pre-propagation GNNs.** We evaluate on three representative PP-GNN baselines—SIGN (Frasca et al., 2020),

*Table 2.* Average node classification results over 10 runs on heterophilic datasets (+: apply robust diffusion operators and HRP to PP-GNN baselines). Accuracy is reported for `roman-empire` and `amazon-ratings`, and ROC AUC is reported for `minesweeper`, `tolokers`, and `questions`.

| | roman -empire | amazon -ratings | mine sweeper | tolo -kers | ques -tions |
|---|---|---|---|---|---|
| GCN | $73.7 \pm 0.7$ | $48.7 \pm 0.6$ | $89.8 \pm 0.5$ | $83.6 \pm 0.7$ | $76.1 \pm 1.3$ |
| SAGE | $85.7 \pm 0.7$ | $53.6 \pm 0.4$ | $93.5 \pm 0.6$ | $82.4 \pm 0.4$ | $76.4 \pm 0.6$ |
| GAT-sep | $88.8 \pm 0.4$ | $52.7 \pm 0.6$ | $\mathbf{93.9 \pm 0.4}$ | $83.8 \pm 0.4$ | $76.8 \pm 0.7$ |
| H2GCN | $60.1 \pm 0.5$ | $36.5 \pm 0.2$ | $89.7 \pm 0.3$ | $73.4 \pm 1.0$ | $63.6 \pm 1.5$ |
| GPRGNN | $64.8 \pm 0.3$ | $44.9 \pm 0.3$ | $86.2 \pm 0.6$ | $72.9 \pm 1.0$ | $55.5 \pm 0.9$ |
| FSGNN | $79.9 \pm 0.6$ | $52.7 \pm 0.8$ | $90.1 \pm 0.7$ | $82.8 \pm 0.6$ | $78.9 \pm 0.9$ |
| ACMGNN | $72.7 \pm 0.8$ | $52.7 \pm 0.3$ | $90.7 \pm 0.6$ | $82.1 \pm 0.6$ | $77.4 \pm 1.5$ |
| GloGNN | $59.6 \pm 0.7$ | $36.9 \pm 0.1$ | $51.1 \pm 1.2$ | $73.4 \pm 1.2$ | $65.7 \pm 1.2$ |
| GGCN | $74.5 \pm 0.5$ | $43.0 \pm 0.3$ | $87.5 \pm 1.2$ | $77.3 \pm 1.1$ | $71.1 \pm 1.6$ |
| O-GNN | $77.7 \pm 0.4$ | $47.3 \pm 0.7$ | $80.6 \pm 1.1$ | $75.6 \pm 1.4$ | $75.1 \pm 1.0$ |
| $G^2$-GNN | $82.2 \pm 0.8$ | $47.9 \pm 0.6$ | $91.8 \pm 0.6$ | $82.5 \pm 0.8$ | $74.8 \pm 0.9$ |
| DIR-GNN | $\mathbf{91.2 \pm 0.3}$ | $47.9 \pm 0.4$ | $87.0 \pm 0.7$ | $81.2 \pm 1.1$ | $76.1 \pm 1.2$ |
| tGNN | $80.0 \pm 0.8$ | $48.2 \pm 0.5$ | $91.9 \pm 0.8$ | $70.8 \pm 1.8$ | $76.4 \pm 1.8$ |
| SIGN | $80.0 \pm 0.5$ | $54.1 \pm 0.7$ | $90.7 \pm 0.6$ | $84.1 \pm 1.0$ | $78.6 \pm 1.1$ |
| HOGA | $79.4 \pm 0.6$ | $51.6 \pm 0.3$ | $90.5 \pm 0.7$ | $78.1 \pm 0.8$ | $78.3 \pm 1.0$ |
| GAMLP | $78.9 \pm 0.7$ | $52.2 \pm 0.4$ | $90.5 \pm 0.7$ | $85.1 \pm 0.8$ | $75.9 \pm 1.3$ |
| GARNN | $79.2 \pm 0.7$ | $53.2 \pm 0.6$ | $90.3 \pm 0.7$ | $82.1 \pm 0.8$ | $78.4 \pm 1.1$ |
| SIGN$^+$ | $81.4 \pm 1.1$ | $\mathbf{54.4 \pm 0.9}$ | $91.3 \pm 0.7$ | $84.6 \pm 1.0$ | $78.4 \pm 1.1$ |
| HOGA$^+$ | $85.5 \pm 1.3$ | $53.8 \pm 0.7$ | $92.1 \pm 0.7$ | $83.6 \pm 0.6$ | $\mathbf{79.0 \pm 0.8}$ |
| GAMLP$^+$ | $83.1 \pm 0.6$ | $52.2 \pm 0.4$ | $92.1 \pm 0.8$ | $\mathbf{85.5 \pm 0.5}$ | $78.2 \pm 1.4$ |
| GARNN$^+$ | $85.5 \pm 0.5$ | $53.6 \pm 0.7$ | $92.3 \pm 0.5$ | $84.0 \pm 0.6$ | $78.5 \pm 1.2$ |

*Table 3.* Average node classification accuracy (%) $\pm$ std over 5 runs on large datasets (+: apply robust diffusion operators and HRP to PP-GNN baselines).

| | ogbn-arxiv | pokec |
|---|---|---|
| GCN | $71.74 \pm 0.29$ | $75.45 \pm 0.17$ |
| SAGE | $71.49 \pm 0.27$ | $78.40 \pm 0.45$ |
| GAT | $72.01 \pm 0.20$ | $81.52 \pm 0.17$ |
| GPRGNN | $71.10 \pm 0.12$ | $78.83 \pm 0.05$ |
| LINKX | $66.18 \pm 0.33$ | $82.04 \pm 0.07$ |
| SIGN | $71.79 \pm 0.14$ | $81.02 \pm 0.33$ |
| HOGA | $72.21 \pm 0.35$ | $82.11 \pm 0.25$ |
| GAMLP | $71.09 \pm 0.36$ | $78.25 \pm 0.69$ |
| GARNN | $72.04 \pm 0.12$ | $80.90 \pm 0.43$ |
| SIGN$^+$ | $72.56 \pm 0.24$ | $83.85 \pm 0.46$ |
| HOGA$^+$ | $72.35 \pm 0.16$ | $84.50 \pm 0.12$ |
| GAMLP$^+$ | $\mathbf{72.62 \pm 0.18}$ | $83.68 \pm 0.58$ |
| GARNN$^+$ | $72.05 \pm 0.37$ | $\mathbf{84.63 \pm 0.08}$ |

*Table 4.* Average node classification accuracy (%) $\pm$ std over 10 runs on homophilic datasets (+: apply robust diffusion operators and HRP to PP-GNN baselines).

| | Computer | Photo | CS | Physics | WikiCS |
|---|---|---|---|---|---|
| GCN | $89.7 \pm 0.5$ | $92.7 \pm 0.2$ | $92.9 \pm 0.1$ | $96.2 \pm 0.1$ | $77.5 \pm 0.9$ |
| SAGE | $91.2 \pm 0.3$ | $94.6 \pm 0.1$ | $93.9 \pm 0.1$ | $96.5 \pm 0.1$ | $74.8 \pm 1.0$ |
| GAT | $90.8 \pm 0.1$ | $93.9 \pm 0.1$ | $93.6 \pm 0.1$ | $96.2 \pm 0.1$ | $76.9 \pm 0.8$ |
| GCNII | $91.0 \pm 0.4$ | $94.3 \pm 0.2$ | $92.2 \pm 0.1$ | $96.0 \pm 0.1$ | $78.7 \pm 0.6$ |
| GPRGNN | $89.3 \pm 0.3$ | $94.5 \pm 0.1$ | $95.1 \pm 0.1$ | $96.9 \pm 0.1$ | $78.1 \pm 0.2$ |
| APPNP | $90.2 \pm 0.2$ | $94.3 \pm 0.1$ | $94.5 \pm 0.1$ | $96.5 \pm 0.1$ | $78.9 \pm 0.1$ |
| PPRGo | $88.7 \pm 0.2$ | $93.6 \pm 0.1$ | $92.5 \pm 0.2$ | $95.5 \pm 0.1$ | $77.9 \pm 0.4$ |
| GGCN | $91.8 \pm 0.2$ | $94.5 \pm 0.1$ | $\mathbf{95.3 \pm 0.1}$ | $\mathbf{97.1 \pm 0.1}$ | $78.4 \pm 0.5$ |
| O-GNN | $92.0 \pm 0.1$ | $95.1 \pm 0.2$ | $95.0 \pm 0.1$ | $97.0 \pm 0.1$ | $79.0 \pm 0.7$ |
| tGNN | $83.4 \pm 1.3$ | $89.9 \pm 0.7$ | $92.9 \pm 0.5$ | $96.2 \pm 0.2$ | $71.5 \pm 1.1$ |
| SIGN | $84.2 \pm 0.3$ | $90.5 \pm 0.7$ | $93.9 \pm 0.4$ | $95.5 \pm 0.2$ | $79.2 \pm 0.7$ |
| HOGA | $84.5 \pm 0.8$ | $91.9 \pm 0.7$ | $94.6 \pm 0.3$ | $95.9 \pm 0.3$ | $78.9 \pm 0.5$ |
| GAMLP | $86.8 \pm 0.3$ | $91.2 \pm 0.9$ | $94.8 \pm 0.3$ | $96.3 \pm 0.2$ | $\mathbf{80.7 \pm 0.7}$ |
| GARNN | $85.3 \pm 0.9$ | $92.1 \pm 0.7$ | $94.5 \pm 0.4$ | $96.0 \pm 0.2$ | $78.9 \pm 0.7$ |
| SIGN$^+$ | $91.7 \pm 0.5$ | $94.3 \pm 0.7$ | $94.7 \pm 0.3$ | $96.1 \pm 0.2$ | $79.4 \pm 0.7$ |
| HOGA$^+$ | $91.4 \pm 0.7$ | $94.2 \pm 0.6$ | $95.1 \pm 0.3$ | $96.6 \pm 0.2$ | $79.5 \pm 0.5$ |
| GAMLP$^+$ | $91.6 \pm 0.6$ | $\mathbf{95.3 \pm 0.6}$ | $94.9 \pm 0.4$ | $96.5 \pm 0.3$ | $80.6 \pm 0.4$ |
| GARNN$^+$ | $\mathbf{92.1 \pm 0.3}$ | $94.8 \pm 0.7$ | $94.9 \pm 0.4$ | $96.9 \pm 0.2$ | $79.7 \pm 0.5$ |

HOGA (Deng et al., 2024), and GAMLP (Zhang et al., 2022)—together with our RNN-based hop aggregator, *graph-augmented RNN* (GARNN). These models span different dense backbones, including MLP-style hop aggregation and attention-based hop aggregation. Our framework applies to each backbone by (i) enabling few-shot hidden-state re-propagation (HRP) across training stages and (ii) replacing the diffusion operator suite used for both feature preprocessing and HRP. For GAMLP, we use GAMLP-R without label inputs by default, unless stated otherwise. Detailed settings are provided in Appendix D and H.

## 5. Evaluation

We evaluate our two contributions—*robust diffusion operators* and *few-shot hidden-state re-propagation (HRP)*—by integrating them into representative pre-propagation PP-GNN baselines and comparing against competitive MP-GNN models. Our evaluation is organized around four questions: (i) do the proposed diffusion families improve PP-GNNs under the same hop budget, (ii) does re-propagation of intermediate representations further narrow the gap to iterative message passing, (iii) how does HRP compare to label-based alternatives such as using labels as input or label propagation, and (iv) what accuracy–efficiency trade-offs do these components introduce? We additionally provide ablations to isolate the effect of each module.

### 5.1. Results on heterophilic graphs

Table 2 and Table 3 report node classification performance on six commonly used heterophilic graphs, including five small graphs and the large `pokec` graph. For each PP-

GNN baseline, results *without* the "+" annotation use standard diffusion operators (either normalized adjacency or random-walk diffusion), while results *with* "+" apply our full method: robust diffusion operators together with HRP. Before applying our techniques, PP-GNNs exhibit a clear accuracy gap to MP-GNNs on challenging datasets such as `roman-empire` and `minesweeper`, reaching up to 11% absolute test accuracy. Our method consistently improves PP-GNNs and reduces this gap substantially—often by roughly half—bringing PP-GNNs close to strong MP-GNN baselines such as GraphSAGE and GAT. On the remaining heterophilic datasets, our method provides additional gains on top of already competitive PP-GNNs, and achieves higher test accuracy than MP-GNNs on 4 out of 6 heterophilic graphs. Overall, across the 6 heterophilic datasets, our full method improves test performance over the corresponding PP-GNN baselines by $+\mathbf{2.18}$ points on average (max $+\mathbf{6.3}$).

*Table 5.* Ablation study (val/test accuracy in %) for different components in SIGN, HOGA, GAMLP, and GARNN on `pokec`.

| Method | Val Acc. (%) | Val ↑ (%) | Test Acc. (%) | Test ↑ (%) |
|---|---|---|---|---|
| SIGN | $81.01 \pm 0.32$ | +0.00 | $81.02 \pm 0.33$ | +0.00 |
| + robust op. | $83.11 \pm 0.05$ | +2.10 | $83.10 \pm 0.05$ | +2.08 |
| + HRP | $83.93 \pm 0.48$ | +0.82 | $83.85 \pm 0.46$ | +0.75 |
| HOGA | $82.11 \pm 0.27$ | +0.00 | $82.11 \pm 0.25$ | +0.00 |
| + robust op. | $82.68 \pm 0.13$ | 0.57 | $82.65 \pm 0.10$ | 0.54 |
| + HRP | $84.54 \pm 0.11$ | 1.86 | $84.50 \pm 0.12$ | 1.85 |
| GAMLP | $78.26 \pm 0.75$ | +0.00 | $78.25 \pm 0.69$ | +0.00 |
| + robust op. | $81.72 \pm 0.42$ | +3.46 | $81.28 \pm 0.51$ | +3.03 |
| + HRP | $83.69 \pm 0.58$ | +1.97 | $83.68 \pm 0.58$ | +2.4 |
| GARNN | $80.92 \pm 0.38$ | +0.00 | $80.90 \pm 0.43$ | +0.00 |
| + robust op. | $83.15 \pm 0.05$ | +2.23 | $83.19 \pm 0.05$ | +2.29 |
| + HRP | $84.66 \pm 0.06$ | +1.51 | $84.63 \pm 0.08$ | +1.44 |

## 5.2. Results on homophilic graphs

Although motivated by heterophily, our techniques also improve performance on homophilic graphs. Table 4 reports results on five small homophilic datasets, and Table 3 additionally includes `ogbn-arxiv`. Compared to heterophilic graphs, the baseline accuracy gap between PP-GNNs and MP-GNNs is smaller, which is consistent with the fact that commonly used diffusion operators (normalized adjacency / random walk) behave largely as low-pass smoothers and thus better match homophily. Nevertheless, we observe substantial gains from our method, with up to $6.8\%$ absolute improvement on `amazon-computer`, and our enhanced PP-GNNs outperform MP-GNNs on 3 out of 6 homophilic datasets. Overall, across the 6 homophilic datasets, our full method improves test accuracy over the corresponding PP-GNN baselines by $+\mathbf{1.96}$ points on average (max $+\mathbf{7.5}$).

## 5.3. Ablating robust operators and HRP

We conduct ablations on the large `pokec` dataset to quantify the contribution of each component (Table 5). Rows labeled "+ robust op." replace the standard diffusion operator with our proposed operator families, while rows labeled "HRP" further enable hidden-state re-propagation. Across four PP-GNN backbones, the robust diffusion operators improve validation/test accuracy by $\mathbf{2.09\%}/\mathbf{1.99\%}$ on average, and HRP provides an additional $\mathbf{1.54\%}/\mathbf{1.61\%}$ gain. These results indicate that (i) richer diffusion bases are beneficial even without iterative refinement, and (ii) re-propagating intermediate representations provides complementary improvements beyond stronger preprocessing alone.

## 5.4. Comparing HRP with label propagation

A natural question is whether hidden-state re-propagation (HRP) simply replicates label-based techniques such as using labels as input or label propagation. We therefore compare HRP against two label strategies on `roman-empire`: (i) *label-as-input*, where we encode training labels as one-hot vectors and mask validation/test labels following standard practice, and (ii) *label propagation / self-training*, fol-

*Table 6.* Label strategy comparison (val/test accuracy in %) for GAMLP and GARNN on `roman-empire`.

| Method | Val Acc. (%) | Test Acc. (%) |
|---|---|---|
| **GAMLP** | | |
| no label | $81.72 \pm 0.42$ | $81.28 \pm 0.51$ |
| - label-as-input | $81.29 \pm 0.71$ | $80.75 \pm 0.66$ |
| - label prop. | $81.98 \pm 0.31$ | $81.53 \pm 0.56$ |
| - HRP | $83.53 \pm 0.63$ | $83.08 \pm 0.60$ |
| **GARNN** | | |
| no label | $82.50 \pm 0.44$ | $82.06 \pm 0.42$ |
| - label-as-input | $82.72 \pm 0.36$ | $82.28 \pm 0.73$ |
| - label prop. | $83.23 \pm 0.37$ | $82.62 \pm 0.40$ |
| - HRP | $86.11 \pm 0.55$ | $85.53 \pm 0.48$ |

*Table 7.* Wall-clock end-to-end training time breakdown for HOGA and GARNN on `pokec`. Stage 1 represents runtime without HRP.

| Method | Stage 1 (sec) | HRP (sec) | Total (sec) | Epoch time (sec) |
|---|---|---|---|---|
| GARNN | 1272 | 51 | 2290 | 2.4 |
| HOGA | 1240 | 94 | 4293 | 8.4 |

lowing the label propagation procedure used in GAMLP, where we augment inputs with soft predictions (logits) from the previous stage under knowledge distillation guidance.

As shown in Table 6, label-as-input and label propagation yield only marginal improvements over the no-label baseline, whereas HRP provides a substantially larger gain for both GAMLP and GARNN. Moreover, on heterophilic graphs, labels are typically less smooth with respect to the graph structure, making propagation-based methods harder to leverage and sometimes even detrimental (e.g., the "label-as-input" variant in Table 6). This motivates HRP: instead of propagating potentially noisy label signals, HRP re-propagates intermediate hidden representations, which (i) are higher-dimensional and can encode richer, task-relevant relational information beyond class probabilities, and (ii) avoid introducing a direct label feature channel that can act as a shortcut to the target.

## 5.5. Training efficiency impact

We next evaluate the training-efficiency impact of the proposed components. We next evaluate the efficiency impact of the proposed methods. Jacobi diffusion has similar preprocessing cost to standard normalized-adjacency/random-walk diffusion, while the Lanczos operator adds a Ritz-mode step but remains comparable to a small number of diffusion steps (Section 3.1). HRP introduces a few extra stages, each requiring only a small number of diffusion operations on hidden states; we observe at most 7 stages overall (typically $< 4$). On `pokec`, Table 7 reports wall-clock end-to-end training time with 4 stages: total time is about $2\times$ the baseline (stage 1), mainly attributed to extra stages introduced. HRP diffusion accounts for only about $2\%$ of the total training time. It also shows that GARNN trains faster than HOGA while achieving similar accuracy, indicating that RNN is a more efficient hop aggregation choice.

Figure 2 further summarizes the accuracy–runtime trade-off

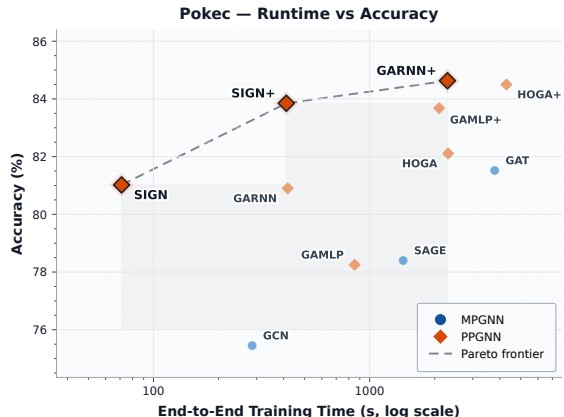

*Figure 2.* Accuracy–runtime trade-off on `pokec`. The x-axis reports end-to-end training time in seconds on a logarithmic scale, and the y-axis reports test accuracy. Enhanced PP-GNN variants with robust diffusion operators and HRP move the PP-GNN family toward a stronger Pareto frontier, achieving substantially higher accuracy than vanilla PP-GNNs and offering a favorable accuracy–time trade-off relative to representative MP-GNN baselines.

on `pokec`. The proposed components, including robust diffusion operators and HRP, shift the PP-GNN family toward stronger Pareto-frontier positions: they substantially improve accuracy over vanilla PP-GNNs while preserving competitive end-to-end training-time trade-offs relative to representative MP-GNN baselines.

### 5.6. Summary

Overall, robust diffusion operators consistently improve PP-GNN performance, and HRP further narrows the gap to iterative message passing. Across both heterophilic and homophilic benchmarks, our enhanced PP-GNNs are competitive with—and often surpass—strong MP-GNN baselines, while preserving the scalability advantages.

## 6. Related Work

**Spectral graph neural networks.** Early *spectral* GNNs design graph convolutions as polynomial filters in the graph Fourier domain. ChebNet (Defferrard et al., 2016) popularized this line by approximating Laplacian spectral filters with truncated Chebyshev polynomials, avoiding explicit eigendecomposition. Unlike PP-GNNs, ChebNet computes these polynomial filters *on-the-fly* as part of the network's forward pass rather than as a preprocessing step. Lanczos-Net (Liao et al., 2019) further leverages Lanczos iterations to construct a *global* low-rank Krylov approximation of the graph operator, yielding a feature-independent basis shared across channels. In contrast to LanczosNet's shared basis, our approach constructs *channel-specific* Krylov subspaces by using each feature channel as the Lanczos starting vector, producing per-channel tridiagonal matrix whose eigendecompositions yield channel-adaptive spectral bases.

**Pre-propagation GNNs.** PP-GNNs precompute multi-hop diffusions and train only a downstream predictor for scalability. Representative models include SGC (Wu et al., 2019), SIGN (Frasca et al., 2020), and adaptive hop aggregators such as GAMLP (Zhang et al., 2022) and HOGA (Deng et al., 2024). Recent analysis suggests expressivity limitations of graph-augmented MLPs (Chen et al., 2020a), motivating our stronger diffusion operators and lightweight re-coupling with representation learning.

**Other scalable GNNs.** Another line of scalable GNNs retains a propagation step but makes it inexpensive. APPNP (Gasteiger et al., 2019a) performs a small number of power iterations that approximate Personalized PageRank, effectively enabling long-range information flow with controlled smoothing. Correct-and-Smooth (C&S) (Huang et al., 2021) improves predictions by diffusing training residuals (*correct*) and then smoothing outputs (*smooth*), showing that simple propagation of model outputs can yield large gains when paired with a strong base predictor.

**Label usage and prediction reuse.** A related line of work leverages labels (or pseudo-labels) as additional signals. UniMP (Shi et al., 2021) injects training labels into the feature space and propagates them through message passing, unifying feature and label propagation. SAGN with self-label enhancement (SLE) augments multi-hop feature models with label propagation and iterative pseudo-label refinement (Sun et al., 2025). Because label-derived features can introduce shortcut learning, these methods often pair propagation with regularization such as masking/dropout on label features or confidence filtering. These approaches are conceptually close to reusing logits as node features; in contrast, our hidden-state re-propagation reuses task-adapted intermediate representations and empirically provides complementary information beyond class-probability signals.

## 7. Conclusion

We study the accuracy limitations of pre-propagation GNNs (PP-GNNs) and propose two scalable remedies. First, we introduce robust diffusion operator families with better-conditioned diffusion bases than normalized-adjacency or random-walk diffusion. Second, we propose few-shot hidden-state re-propagation (HRP), which occasionally diffuses intermediate representations to inject task-adapted signals without reverting to fully iterative message passing. Across heterophilic and homophilic benchmarks, these components consistently improve PP-GNNs, narrowing the gap to competitive MP-GNNs and often matching or exceeding their accuracy, while maintaining training efficiency.

## Acknowledgments

This work was supported in part by ACE, one of the seven centers in JUMP 2.0, a Semiconductor Research Corporation (SRC) program sponsored by DARPA and NSF Awards #2212371 and #2403135, and a research gift from Qualcomm.

## Impact Statement

This work advances scalable graph representation learning by improving the accuracy of pre-propagation GNNs while preserving their efficiency benefits. As a general-purpose modeling and optimization technique, our methods may enable more effective use of graph-structured data in applications such as recommender systems, fraud detection, social and information network analysis, and scientific discovery, potentially reducing computational cost and energy consumption for large-scale graph learning.

At the same time, these applications can raise well-known ethical concerns. Improved performance on social or user-interaction graphs may amplify existing biases present in the data, impact fairness for underrepresented groups, or enable more effective profiling and targeted persuasion if deployed without safeguards. The proposed techniques do not introduce new data modalities or directly increase model access to sensitive attributes, but they also do not inherently prevent harmful downstream uses. Practitioners should therefore follow standard responsible-ML practices, including careful dataset governance, privacy protection, bias and fairness evaluation, and appropriate human oversight when deploying graph models in high-stakes settings.

We do not anticipate direct negative societal consequences beyond those already associated with learning on large graphs; nevertheless, we encourage future work on integrating fairness, privacy, and robustness constraints into scalable graph representation learning pipelines and on auditing the behavior of graph models under distribution shift and adversarial manipulation.

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

*Table 8.* Dataset statistics. $N$ and $E$ denote the number of nodes and edges (undirected edges counted once; directed graphs count directed edges). Split: "fixed" denotes the standard public split; "random" denotes averages over multiple random splits.

| Dataset | $N$ | $E$ | Feat. dim | #Classes | Train/Val/Test | Split | Type |
|---|---|---|---|---|---|---|---|
| roman-empire | 22,662 | 32,927 | 300 | 18 | 50/25/25 | fixed | Hetero |
| amazon-ratings | 24,492 | 93,050 | 300 | 5 | 50/25/25 | fixed | Hetero |
| minesweeper | 10,000 | 39,402 | 7 | 2 | 50/25/25 | fixed | Hetero |
| tolokers | 11,758 | 519,000 | 10 | 2 | 50/25/25 | fixed | Hetero |
| questions | 48,921 | 153,540 | 301 | 2 | 50/25/25 | fixed | Hetero |
| pokec | 1,632,803 | 30,622,564 | 65 | 2 | 50/25/25 | fixed | Hetero |
| amazon-photo | 7,650 | 119,081 | 745 | 8 | 60/20/20 | random | Homo |
| amazon-computer | 13,752 | 245,861 | 767 | 10 | 60/20/20 | random | Homo |
| coauthor-cs | 18,333 | 81,894 | 6,805 | 15 | 60/20/20 | random | Homo |
| coauthor-physics | 34,493 | 247,962 | 8,415 | 5 | 60/20/20 | random | Homo |
| wikics | 11,701 | 216,123 | 300 | 10 | 5/45/50 | fixed | Homo |
| ogbn-arxiv | 169,343 | 1,166,243 | 128 | 40 | 53.7/17.6/28.7 | fixed | Homo |

*Table 9.* Message-passing GNN baselines and the original papers that introduced them.

| Baseline | Original paper |
|---|---|
| GCN | (Kipf & Welling, 2017) |
| GAT (incl. GAT-sep) | (Veličković et al., 2018) |
| GraphSAGE | (Hamilton et al., 2017) |
| APPNP | (Gasteiger et al., 2019a) |
| GPRGNN | (Chien et al., 2021) |
| PPRGo | (Bojchevski et al., 2020) |
| GCNII | (Chen et al., 2020c) |
| H2GCN | (Zhu et al., 2020) |
| FSGNN | (Maurya et al., 2022) |
| GloGNN | (Li et al., 2022) |
| GGCN | (Yan et al., 2022) |
| O-GNN | (Song et al., 2023) |
| tGNN (a.k.a. tGCN) | (Hua et al., 2022) |
| $G^2$-GNN | (Rusch et al., 2023) |
| DIR-GNN | (Rossi et al., 2024) |
| LINKX | (Lim et al., 2021) |

## A. Dataset details

The 12 datasets used in our experiments are summarized in Table 8.

**Split protocol.** We follow the standard split protocol for each dataset. For the four homophilic datasets amazon-comput er, amazon-photo, coauthor-cs, and coauthor-physics, we report mean ± standard deviation over random splits. All other datasets use their fixed public splits.

## B. Hardware settings

For the training efficiency study, we use a Linux server with an NVIDIA A100 GPU (80GB memory, CUDA 12.6). Our training pipeline is built on the scalable PP-GNN framework of Yue et al. (2025).

## C. MP-GNN baseline details

We list the MP-GNN baselines used in our experiments in Table 9.

We use the accuracy numbers reported in the original baseline papers and the benchmarking study of Platonov et al. (2023). For baselines without publicly available results on a given dataset, we tune hyperparameters using the search spaces in Table 10.

*Table 10.* Hyperparameter tuning settings for baseline models without publicly available results on given datasets.

| Model | Fixed hyperparameters | Tuned hyperparameters (search space) |
|---|---|---|
| GCNII | Hidden dim 512; LR 0.001; epochs 2000 | Layers $\{5, 10\}$; dropout $\{0.3, 0.5, 0.7\}$; $\alpha \in \{0.3, 0.5, 0.7\}$; $\theta \in \{0.5, 1.0\}$ |
| GGCN | Hidden dim 512; LR 0.001; epochs 2000 | Layers $\{5, 10\}$; dropout $\{0.3, 0.5, 0.7\}$; decay rate $\eta \in \{0.5, 1.0, 1.5\}$; exponent $\{2, 3\}$ |
| OrderedGNN | Hidden dim 512; LR 0.001; epochs 2000 | Layers $\{5, 10\}$; dropout $\{0.3, 0.5, 0.7\}$; chunk size $\{4, 16, 64\}$ |
| tGCN | Hidden dim 512; LR 0.001; epochs 2000 | Layers $\{2, 3\}$; dropout $\{0.3, 0.5, 0.7\}$; rank $\{256, 512\}$ |
| $G^2$-GNN | Hidden dim 512; LR 0.001; epochs 2000 | Layers $\{5, 10\}$; dropout $\{0.3, 0.5, 0.7\}$; exponent $p \in \{2, 3, 4\}$ |
| DIR-GNN | Hidden dim 512; LR 0.001; epochs 2000; $\alpha = 0.5$; GATConv + jumping knowledge ("max") | Layers $\{3, 5\}$; dropout $\{0.3, 0.5, 0.7\}$ |

## D. PP-GNN baseline details

To isolate the benefit of stronger diffusion operators *under the same hop budget*, we fix the maximum hop budget to 15 across methods and datasets. We then use an automatic tuner to select the effective hop count, since the optimal number of hops can vary across operators and datasets due to oversmoothing and related effects; in practice, we typically find that fewer than 10 hops suffice. The hyperparameter ranges for each PP-GNN model are listed in Table 11. Unless otherwise stated, baseline PP-GNNs use two standard diffusions: normalized adjacency and random-walk diffusion. To support HRP, we augment each PP-GNN backbone with an additional MLP layer before the final output projection, so that the hidden representations selected for re-propagation match the dimensionality of the original node features and can be blended directly.

## E. Spectrum-aware calibration for Jacobi bases.

Let $L$ be the normalized graph Laplacian with spectrum $\lambda \in [0, 2]$, and define the rescaled operator

$$\tilde{L} = L - I,$$

whose eigenvalues $\tilde{\lambda} \in [-1, 1]$.

To obtain a coarse estimate of the spectral density of $\tilde{L}$ without eigendecomposition, we employ a stochastic Chebyshev moment estimator. We draw $R$ random probe vectors $z_r \sim \mathcal{N}(0, I)$ and compute

$$v_0^{(r)} = z_r, \qquad v_1^{(r)} = \tilde{L}z_r, \qquad v_{k+1}^{(r)} = 2\tilde{L}v_k^{(r)} - v_{k-1}^{(r)}.$$

The Chebyshev moments are estimated as

$$m_k \approx \frac{1}{R}\sum_{r=1}^{R}\langle z_r, v_k^{(r)}\rangle \approx \mathrm{Tr}\Big(T_k(\tilde{L})\Big), \qquad k = 0, \ldots, K_{\mathrm{spec}}.$$

From $\{m_k\}$, we reconstruct a coarse spectral density $\rho(\tilde{\lambda})$ on $[-1, 1]$ by matching moments on a fixed grid. We then compute the low- and high-frequency spectral masses

$$M_{\mathrm{low}} = \int_{-1}^{0} \rho(\tilde{\lambda})\, d\tilde{\lambda}, \qquad M_{\mathrm{high}} = \int_{0}^{1} \rho(\tilde{\lambda})\, d\tilde{\lambda},$$

and define the spectral imbalance

$$\delta = \frac{M_{\mathrm{high}} - M_{\mathrm{low}}}{M_{\mathrm{high}} + M_{\mathrm{low}}} \in [-1, 1].$$

We use $\delta$ to select the parameters of the Jacobi polynomial basis. Recall that Jacobi polynomials are orthogonal on $[-1, 1]$ with respect to the weight

$$w_{\alpha,\beta}(z) = (1 - z)^\alpha (1 + z)^\beta.$$

We set

$$\alpha = \gamma\min(-\delta, 0), \qquad \beta = \gamma\min(\delta, 0),$$

where $\gamma > 0$ controls the strength of the bias.

This choice biases the polynomial basis toward regions of the spectrum with greater mass: $\beta < 0$ increases local resolution near $\tilde{\lambda} = -1$ (low Laplacian frequencies), while $\alpha < 0$ increases resolution near $\tilde{\lambda} = +1$ (high frequencies). If $\delta \approx 0$, the method reduces to the Legendre basis ($\alpha = \beta = 0$). The calibrated $(\alpha, \beta)$ are fixed during preprocessing and reused across all PP-GNN training stages.

# F. Optional auto-search details

**Motivation.** Two factors affect the effectiveness of HRP: (i) the stage-wise blending schedule, and (ii) the choice of checkpoint used to generate the re-propagated representations for the next stage. Since re-propagation is applied outside the gradient path, the checkpoint with the best validation accuracy at stage $s$ is not always the one that yields the best re-propagated features for stage $s+1$.

**Blending-weight tuning.** When enabled, we use a small-budget covariance matrix adaptation evolution strategy (CMA-ES) to tune the per-hop blending weights $\{\alpha_s\}$ at each stage. This search is performed with a small epoch budget and is disabled by default when computational resources are limited.

**Checkpoint and operator selection.** When enabled, we build a candidate set of epochs $\mathcal{E}_s$ that includes both early-epoch checkpoints and high-performing checkpoints. We then select the best pair $(e, \Psi)$ using a two-stage screening procedure: a tight budget with a small hop count (e.g., 2) for rapid filtering, followed by a larger budget for the finalist.

**Spectral diversity score.** To diversify candidates, we score how *spectrally different* a hidden state is from the original input using per-channel spectral moments of the normalized Laplacian $L$. For a feature matrix $X \in \mathbb{R}^{N \times d}$, let $X_c \in \mathbb{R}^N$ denote the $c$-th channel. We define the (normalized) $k$-th moment signature

$$\mu_k(X_c) = \frac{\|L^k X_c\|_2^2}{\|X_c\|_2^2}, \qquad \mathbf{m}(X_c) = \mathrm{norm}\big([\mu_0(X_c), \ldots, \mu_K(X_c)]\big), \tag{23}$$

where $\mathrm{norm}(\cdot)$ normalizes the moment vector (e.g., to unit sum). We compare two feature matrices by averaging a divergence between channel-wise signatures:

$$\mathrm{Dist}(X, Y) = \frac{1}{d} \sum_{c=1}^{d} D(\mathbf{m}(X_c), \mathbf{m}(Y_c)), \tag{24}$$

with $D(\cdot, \cdot)$ a simple divergence (e.g., $\ell_1$ or Jensen–Shannon). Each arm in the screening procedure corresponds to a candidate epoch and diffusion choice $(e, \Psi)$; we prioritize candidates that are both high-performing (train/validation) and spectrally diverse relative to the input features. This step is executed only at stage boundaries; for very large graphs, we disable it by default.

**Experimental setting and overhead.** In our experiments, we first hand-tune HRP and enable the optional auto-search only when manual tuning does not already provide a substantive gain. In particular, the reported results on `physics` and `roman-empire` are obtained without CMA-ES; we also do not use CMA-ES on the two large graphs, `pokec` and `ogbn-arxiv`. On the remaining eight small datasets where CMA-ES is applied, we measure the overhead of the full optional tuning pipeline, including CMA-ES, checkpoint/operator screening, and moment-signature scoring, relative to fixed-configuration runs without auto-search. Averaged over the four PP-GNN backbones, the tuning overhead is $2.83\times$ on the homophilic datasets and $8.78\times$ on the heterophilic datasets. The additional memory cost is modest: the average increase in peak training memory is about 0.3 GB, with a maximum increase of 2 GB. These costs should therefore be interpreted as optional model-selection overhead rather than the intrinsic per-epoch cost of HRP.

# G. Hyperparameter Sensitivity of HRP

To assess the hyperparameter sensitivity of HRP, we evaluate a fixed-configuration variant across all 12 datasets and four PP-GNN backbones. In this setting, HRP uses a fixed cosine blending schedule, without manual tuning of HRP hyperparameters or optional auto-search. Relative to the fully tuned configuration, this fixed variant incurs only a modest drop in average test performance: 0.38 points on heterophilic datasets, 0.31 points on homophilic datasets, and 0.34 points overall.

Importantly, even without tuning, HRP still provides a clear improvement over using robust diffusion operators alone, with an average gain of 0.51 points. Manual tuning or CMA-ES recovers an additional 0.34 points on average. These results suggest that the primary gain comes from the HRP mechanism itself, while optional tuning mainly serves as a lightweight convenience tool for extracting a smaller additional improvement.

## H. Diffusion Operator Choices

Table 12 lists the diffusion-operator choices used for the results in Section 5. For each dataset and PP-GNN backbone, we report the operator used for the initial preprocessing stage and the operator used during HRP. These choices define the fixed method configuration for each model–dataset pair.

The same operator choices are used in the non-CMA-ES experiments. In that setting, we disable the optional auto-search and use the fixed cosine blending schedule described in Appendix G; only the HRP blending and stage-selection procedure changes, while the preprocessing and HRP diffusion operators remain unchanged.

## I. Spectral Diagnostics of Krylov Preprocessing

We provide qualitative diagnostics to examine what spectral components are exposed by the Krylov preprocessing and how the downstream PP-GNN backbone uses them. For each input channel $c$, the Lanczos procedure produces Ritz values $\lambda_{c,i}$ and corresponding Ritz components $z_{c,i}$, as described in Section 3.1. Since $\tilde{L} = L - I$ has spectrum in $[-1, 1]$, larger Ritz values correspond to higher-frequency components of the normalized Laplacian. Thus, the distribution of Ritz values provides a compact view of the spectral content made available to the downstream model.

We visualize two complementary quantities in Figure 3. First, the weighted Ritz-value maps show the components exposed by preprocessing. Each point corresponds to a channel-specific Ritz component, with vertical position given by its Ritz value $\lambda_{c,i}$ and color indicating its normalized basis weight. Second, the HOGA attention maps show how the downstream PP-GNN backbone uses the precomputed bank. We report attention scores at the best validation epoch, with rows corresponding to validation nodes and columns corresponding to Ritz indices.

On the heterophilic graph `pokec`, the weighted Ritz-value map places noticeable mass near the upper spectral range, indicating that Krylov preprocessing exposes substantial non-low-pass components. The corresponding HOGA attention map is also spread across a broader range of Ritz indices, suggesting that the downstream backbone does not rely only on early or low-pass components. In contrast, on the more homophilic `amazon-computer` graph, the exposed Ritz components are less concentrated at the high-frequency end, and the HOGA attention is more concentrated on early Ritz indices, which is consistent with a stronger low-pass preference.

These diagnostics should be interpreted at the level of the precomputed diffusion bank. The Krylov components have spectral meaning because each Ritz component is associated with a Ritz value, but the downstream PP-GNN backbone remains a nonlinear node-domain aggregator over the precomputed bank. Therefore, Figure 3 is not an exact spectral decomposition of the final predictor; rather, it illustrates which spectrally meaningful inputs are exposed by preprocessing and emphasized by the PP-GNN backbone.

*Table 11.* Optuna hyperparameter search spaces for SIGN, HOGA, GAMLP, and GARNN (model-specific and shared optimization parameters).

| Model | Hyperparameter | Search space (as implemented) |
|---|---|---|
| **HOGA** | lr | LogUniform$(10^{-4}, 5 \times 10^{-2})$ |
| **HOGA** | weight_decay | LogUniform$(10^{-7}, 5 \times 10^{-3})$ |
| **HOGA** | hidden_channels | Categorical$\{128, 192, 256, 320, 384, 448, 512, 640\}$ |
| **HOGA** | mlp_hidden | Categorical$\{128, 256, 320, 384, 512, 640\}$ |
| **HOGA** | mlp_dropout | Uniform$(0.0, 0.6)$ |
| **HOGA** | dropout | Uniform$(0.0, 0.6)$ |
| **HOGA** | input_dropout | Uniform$(0.0, 0.5)$ |
| **HOGA** | mlplayers | Int$[1, 4]$ |
| **HOGA** | num_layers | Int$[1, 4]$ |
| **HOGA** | num_heads | Categorical$\{1, 2, 4, 8\}$ |
| **HOGA** | attn_dropout | Uniform$(0.0, 0.5)$ |
| **HOGA** | use_post_res | Categorical$\{0, 1\}$ |
| **GARNN** | lr | LogUniform$(10^{-4}, 5 \times 10^{-2})$ |
| **GARNN** | weight_decay | LogUniform$(10^{-7}, 5 \times 10^{-3})$ |
| **GARNN** | hidden_channels | Categorical$\{128, 192, 256, 320, 384, 448, 512, 640\}$ |
| **GARNN** | mlp_hidden | Categorical$\{128, 256, 320, 384, 512, 640\}$ |
| **GARNN** | mlp_dropout | Uniform$(0.0, 0.6)$ |
| **GARNN** | dropout | Uniform$(0.0, 0.6)$ |
| **GARNN** | input_dropout | Uniform$(0.0, 0.5)$ |
| **GARNN** | mlplayers | Int$[1, 4]$ |
| **GARNN** | rnn_type | Categorical$\{$LSTM, GRU$\}$[1] |
| **GARNN** | rnn_dim | Categorical$\{128, 192, 256, 320, 384, 512\}$ |
| **GARNN** | num_layers | Int$[1, 3]$ |
| **GARNN** | rnn_bidirectional | Categorical$\{$False, True$\}$ |
| **GARNN** | rnn_readout | Categorical$\{$rnn_out, hidden$\}$ |
| **GARNN** | use_post_res | Categorical$\{0, 1\}$ |
| **GARNN** | use_bn | Categorical$\{$False, True$\}$ |
| **GARNN** | shared_mlp | Categorical$\{$False, True$\}$ |
| **GAMLP** | lr | LogUniform$(10^{-4}, 5 \times 10^{-2})$ |
| **GAMLP** | weight_decay | LogUniform$(10^{-7}, 5 \times 10^{-3})$ |
| **GAMLP** | hidden_channels | Categorical$\{128, 192, 256, 320, 384, 448, 512, 640\}$ |
| **GAMLP** | mlp_hidden | Categorical$\{128, 256, 320, 384, 512, 640\}$ |
| **GAMLP** | mlp_dropout | Uniform$(0.0, 0.6)$ |
| **GAMLP** | dropout | Uniform$(0.0, 0.6)$ |
| **GAMLP** | input_dropout | Uniform$(0.0, 0.5)$ |
| **GAMLP** | mlplayers | Int$[1, 4]$ |
| **GAMLP** | gamlp_hidden | Deterministic: gamlp_hidden $\leftarrow$ hidden_channels |
| **GAMLP** | gamlp_alpha | Uniform$(0.1, 0.9)$ |
| **GAMLP** | gamlp_n_layers_1 | Int$[2, 5]$ |
| **GAMLP** | gamlp_n_layers_2 | Int$[2, 5]$ |
| **GAMLP** | gamlp_input_drop | Uniform$(0.0, 0.6)$ |
| **GAMLP** | gamlp_att_drop | Uniform$(0.0, 0.6)$ |
| **GAMLP** | gamlp_act | Categorical$\{$relu, leaky_relu, sigmoid$\}$ |
| **GAMLP** | gamlp_pre_process | Categorical$\{$False, True$\}$ |
| **GAMLP** | gamlp_residual | Categorical$\{$False, True$\}$ |
| **GAMLP** | gamlp_pre_dropout | Categorical$\{$False, True$\}$ |
| **GAMLP** | gamlp_bns | Categorical$\{$False, True$\}$ |
| **SIGN** | lr | LogUniform$(10^{-4}, 5 \times 10^{-2})$ |
| **SIGN** | weight_decay | LogUniform$(10^{-7}, 5 \times 10^{-3})$ |
| **SIGN** | hidden_channels | Categorical$\{128, 256, 384, 512, 640, 768\}$ |
| **SIGN** | dropout | Uniform$(0.0, 0.7)$ |
| **SIGN** | input_dropout | Uniform$(0.0, 0.5)$ |
| **SIGN** | num_layers | Int$[2, 5]$ |

*Table 12.* Selected diffusion-operator-model pairs on different datasets used in Section 5. Each row lists the preprocessing operator and the operator used for HRP for a given backbone–dataset pair. DAD denotes the symmetric normalized adjacency $D^{-1/2}AD^{-1/2}$, DA denotes the left-normalized diffusion $D^{-1}A$, Cheb and Leg denote Chebyshev and Legendre polynomial bases, Jac denotes the calibrated Jacobi basis, and Kry denotes the Lanczos/Krylov operator.

| Model | Dataset | Pre-process | HRP | Model | Dataset | Pre-process | HRP |
|---|---|---|---|---|---|---|---|
| HOGA | amazon-computer | Leg | Jac | HOGA | ogbn-arxiv | DAD | Jac |
| GARNN | amazon-computer | Leg | Jac | GARNN | ogbn-arxiv | DAD | Jac |
| GAMLP | amazon-computer | Leg | Jac | GAMLP | ogbn-arxiv | Kry | Jac |
| SIGN | amazon-computer | Leg | Jac | SIGN | ogbn-arxiv | DA | Jac |
| HOGA | amazon-photo | Leg | Jac | HOGA | pokec | Kry | Jac |
| GARNN | amazon-photo | Leg | Jac | GARNN | pokec | Kry | Jac |
| GAMLP | amazon-photo | Leg | Jac | GAMLP | pokec | DAD | Jac |
| SIGN | amazon-photo | Leg | Jac | SIGN | pokec | Kry | Jac |
| HOGA | amazon-ratings | DAD | Jac | HOGA | questions | DAD | Jac |
| GARNN | amazon-ratings | DA | Jac | GARNN | questions | Kry | Jac |
| GAMLP | amazon-ratings | DA | Jac | GAMLP | questions | DAD | Jac |
| SIGN | amazon-ratings | DA | Jac | SIGN | questions | DAD | Jac |
| HOGA | coauthor-cs | Leg | Kry | HOGA | roman-empire | Jac | Jac |
| GARNN | coauthor-cs | DAD | Jac | GARNN | roman-empire | Cheb | Jac |
| GAMLP | coauthor-cs | Leg | Jac | GAMLP | roman-empire | Leg | Jac |
| SIGN | coauthor-cs | Leg | Kry | SIGN | roman-empire | Jac | Jac |
| HOGA | coauthor-physics | DAD | Jac | HOGA | tolokers | Kry | Jac |
| GARNN | coauthor-physics | DAD | Jac | GARNN | tolokers | Kry | Jac |
| GAMLP | coauthor-physics | DAD | Jac | GAMLP | tolokers | DAD | Jac |
| SIGN | coauthor-physics | DAD | Jac | SIGN | tolokers | Jac | Jac |
| HOGA | minesweeper | Cheb | Jac | HOGA | wikics | DAD | Jac |
| GARNN | minesweeper | Cheb | Jac | GARNN | wikics | DA | Jac |
| GAMLP | minesweeper | Cheb | Jac | GAMLP | wikics | DA | Jac |
| SIGN | minesweeper | Jac | Jac | SIGN | wikics | DA | Jac |

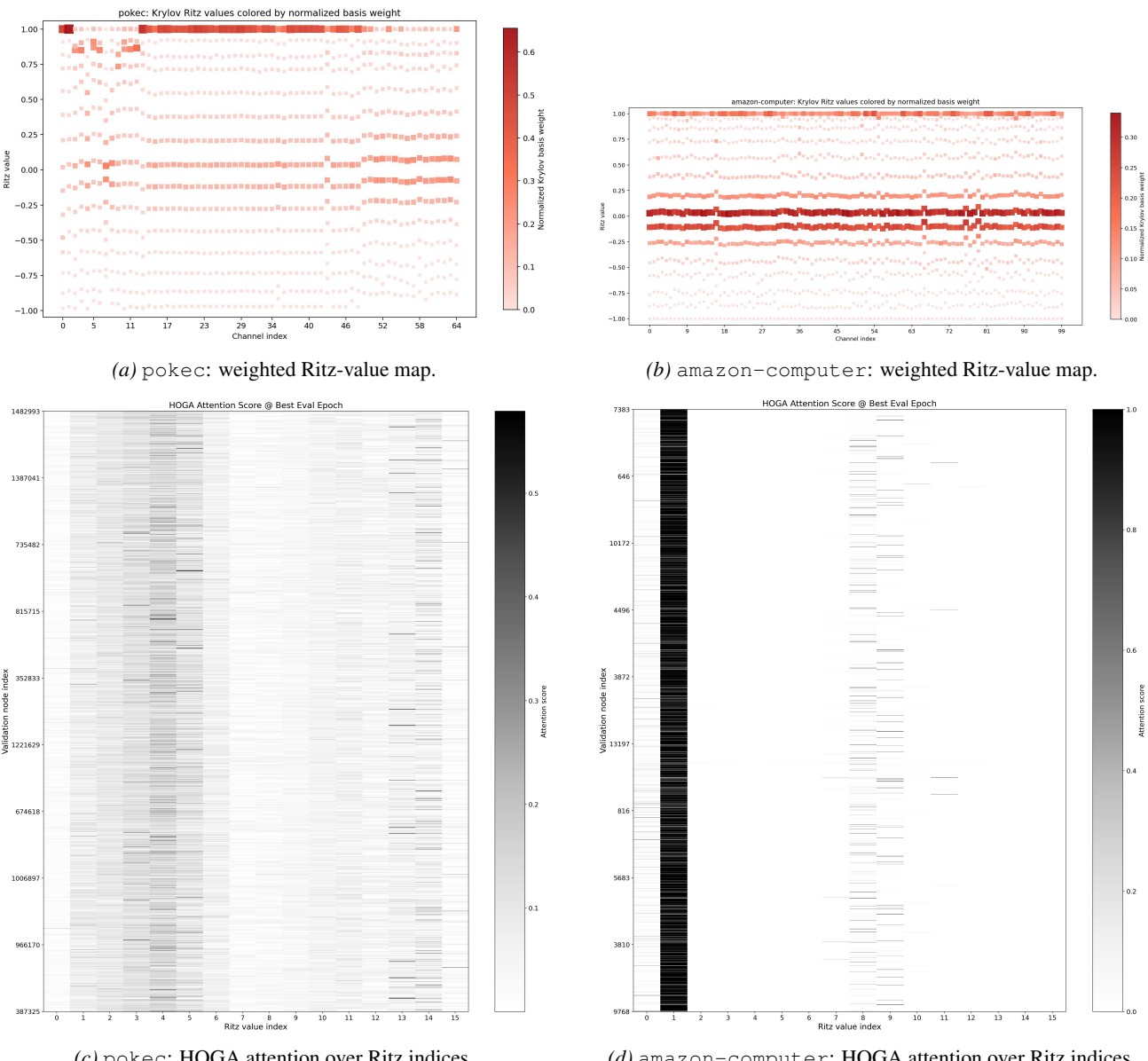

*(a)* `pokec`: weighted Ritz-value map.

*(b)* `amazon-computer`: weighted Ritz-value map.

*(c)* `pokec`: HOGA attention over Ritz indices.

*(d)* `amazon-computer`: HOGA attention over Ritz indices.

*Figure 3.* Spectral diagnostics of Krylov preprocessing on a heterophilic graph, `pokec`, and a more homophilic graph, `amazon-computer`. The top row shows weighted Ritz-value maps: each point corresponds to a channel-specific Ritz component, the vertical axis gives its Ritz value $\lambda_{c,i}$, and color indicates the normalized basis weight. The bottom row shows HOGA attention scores over Ritz indices at the best validation epoch. On `pokec`, preprocessing exposes more upper-spectrum components and the downstream attention is distributed across a broader range of Ritz indices. On `amazon-computer`, attention is more concentrated on early Ritz indices, consistent with a stronger low-pass preference. These plots characterize the spectrally meaningful inputs exposed to and emphasized by the PP-GNN backbone, rather than an exact spectral decomposition of the final nonlinear predictor.

