# OpenReview forum: "Revisiting Pre-Propagation GNNs: Robust Diffusion Operators and Hidden-State Re-Propagation"
_ICML.cc/2026/Conference — ICML 2026 regular_

### Official Review · Reviewer_rbyQ · 2026-03-09

**Soundness:** 3
**Presentation:** 3
**Significance:** 2
**Originality:** 2
**Overall Recommendation:** 4
**Confidence:** 4

**Summary:**

This paper addresses the expressivity and accuracy limitations of PP-GNNs by robust diffusion operators, hidden-state re-propagation and RNN-based hop aggregator

**Compliance With Llm Reviewing Policy:**

Affirmed.

**Key Questions For Authors:**

1. Do the results reported in Tables 2, 3, 4 rely on auto-search? If so, could you quantify the true computational and memory overhead of running CMA-ES and calculating the channel-wise moment signatures, and clarify if this was included in the Table 7 wall-clock breakdown?

2. How does your proposed method compare with some SOTA adaptive-channel methods, such as ACMGNN [1] and FSGNN[2].

3. Need more results on real challenging heterophily datasets, such as the malignant and ambiguous heterophilic datasets identified in[3].


[1] Revisiting heterophily for graph neural networks. Advances in neural information processing systems. 2022 Dec 6;35:1362-75.

[2] Simplifying approach to node classification in graph neural networks. Journal of Computational Science, 62, 101695.

[3] The heterophilic graph learning handbook: Benchmarks, models, theoretical analysis, applications and challenges. arXiv preprint arXiv:2407.09618. 2024 Jul 12.

**Limitations:**

yes

**Strengths And Weaknesses:**

## Soundness

Comprehensive empirical evaluation

Thorough Ablation Studies

Methodological Rigor and Efficiency

Complexity of Auto-Search is not rigorously quantified

## Presentation

Clear Narrative and Motivation

## Significance

Tackle a major bottleneck in graph machine learning


## Originality

The authors make some interesting combination of existing methods

---

> ### Author Rebuttal · Authors · 2026-03-31
>
> We thank the reviewer for the thoughtful and encouraging assessment, and especially for the constructive questions on the optional auto-search overhead, adaptive-channel baselines, and additional heterophily benchmarks.
>
> **Q1. CMA-ES overhead** Regarding Tables 2–4, these results report the final method (robust diffusion operators + HRP), but they should not be interpreted as implying that CMA-ES was used on every dataset. In our pipeline, CMA-ES is only an optional outer-loop tuner for the stage-wise blending weights, while the channel-wise moment signatures are only used in the optional candidate-screening step at stage boundaries. Therefore, these costs should be viewed as hyperparameter/model-selection overhead rather than the intrinsic per-epoch overhead of HRP itself. In practice, we first hand-tuned HRP and only enabled the optional auto-search on the 10 small datasets when manual tuning did not already yield a substantive gain. In particular, the reported results on Physics and Roman-empire were obtained without CMA-ES, and we likewise did not use CMA-ES on Pokec or Ogbn-arxiv. Accordingly, Table 7 should be read as a fixed-configuration wall-clock breakdown after the configuration has already been chosen; it does not include optional CMA-ES or moment-signature overhead. In that fixed-configuration setting, HRP diffusion step accounts for only about 2% of total training time.
>
> For the 10 small datasets, the measured overhead corresponds to the combined optional tuning pipeline—CMA-ES, checkpoint/operator screening, and moment-signature scoring—relative to fixed-configuration runs without auto-search. Averaged over the four PP-GNN backbones, this one-time overhead is 2.83× on the homophilic datasets and 8.78× on the heterophilic datasets. The peak memory overhead is modest: across the 40 small-dataset runs, the average increase in peak training memory is about 0.3 GB, and the maximum increase is 2 GB. Since we did not separately instrument CMA-ES and moment-signature scoring, we prefer to report these as the overhead of the combined optional tuning procedure rather than give an artificial per-component split. Importantly, a no-CMA-ES ablation with fixed blending factors causes only a modest drop in average test performance: 0.38 points on heterophilic datasets, 0.31 points on homophilic datasets, and 0.34 points overall. Moreover, HRP without CMA-ES still improves over robust operators alone by 0.51 points on average, while hand tuning or CMA-ES recovers an additional 0.34 points. This suggests that the main benefit comes from HRP itself, while CMA-ES mainly serves as a lightweight convenience tuner for extracting a smaller additional gain.
>
> **Q2. Comparison to adaptive-channel methods** Regarding adaptive-channel baselines, FSGNN is already included in Table 2. Following the reviewer’s suggestion, we additionally checked ACM-GCN on the five small heterophilic benchmarks. ACM-GCN obtains 72.7 / 52.7 / 90.7 / 82.1 / 77.4 on roman-empire / amazon-ratings / minesweeper / tolokers / questions, while the best enhanced PP-GNN in our framework outperforms both ACM-GCN and FSGNN on all five datasets. We will make this comparison explicit in the revision for completeness.
>
> **Q3. Additional benchmarks** We also agree that evaluation on the malignant and ambiguous heterophilic benchmarks highlighted in the recent handbook would further strengthen the paper. Due to the limited rebuttal-period time and compute budget, we have not yet completed adding additional graphs, so we prefer not to claim results that we have not thoroughly validated. We will add these datasets in the revision and view this as a valuable direction for strengthening the empirical coverage.

---

> > ### Author Rebuttal · Reviewer_rbyQ · 2026-04-04
> >
> > Thanks for the rebuttal. I have no question left.

---

> > > ### Author Response · Authors · 2026-04-04
> > >
> > > Thank you for the acknowledgement. We appreciate your positive follow-up and are glad the rebuttal fully addressed your concerns.

---

### Official Review · Reviewer_Xn5G · 2026-03-09

**Soundness:** 3
**Presentation:** 3
**Significance:** 3
**Originality:** 3
**Overall Recommendation:** 5
**Confidence:** 3

**Summary:**

This paper improves pre-propagation GNNs on heterophilic graphs through more robust graph diffusion operators and a hidden-state re-propagation scheme.

**Compliance With Llm Reviewing Policy:**

Affirmed.

**Final Justification:**

The authors have solved my concerns. Therefore, I keep my original positive score.

**Key Questions For Authors:**

**Question 1**
 Theoretically, why does the Lanczos/Krylov operator improve the performance of PP-GNNs?

**Question 2**
Can you provide a direct time comparison among the PP-GNNs, MPNNs, and your method?

**Question 3**
Are the proposed graph diffusion operators specific to PP-GNNs, or it also improve the performances in MPNNs? Can you show the performance comparison between MPNNs with the proposed operators and MPNNs with other spectral filters (e.g., those used in BernNet [2] and GPR-GNN [3])? I am just curious about the application of this operator, and the answer to the question will not affect my judgment of this paper.

[2]. He, Mingguo, Zhewei Wei, and Hongteng Xu. "Bernnet: Learning arbitrary graph spectral filters via bernstein approximation." NeurIPS, 2021.

[3].Chien, Eli, et al. "Adaptive universal generalized pagerank graph neural network." ICLR, 2021.

**Limitations:**

See Questions.

**Strengths And Weaknesses:**

**Strengths**
- This paper is well-organized, with reasonable motivations and effective solutions.
- The proposed method helps solve the problem of PP-GNNs on heterophilic graphs, which are widely observed in real life.
- The methods are introduced in detail.


**Weakness**
- Insufficient theoretical explanations. Theoretically, why does the Lanczos/Krylov operator improve the performance of PP-GNNs?
- Lack of direct illustration of efficiency. The advantages of PP-GNNs over MPNNs are their efficiency. The complicated designs of your method seem to improve the training and inference time. Can you provide a direct time comparison among the PP-GNNs, MPNNs, and your method? It's a good way to know the time-performance trade-off.
- Need clearer statements on the originality. In the introduction, it claims "We propose Jacobi-polynomial and Krylov-subspace-
based diffusion operators". How is the Jacobi polynomial different from the JacobiConv proposed in [1]? I think it should be explicitly noted which part is original and which comes from previous works.

[1]. Wang, Xiyuan, and Muhan Zhang. "How powerful are spectral graph neural networks." ICML 2022.

---

> ### Author Rebuttal · Authors · 2026-03-31
>
> We thank the reviewer for recognizing the paper’s organization, motivation, and practical value. We also appreciate the three important questions and address them below.
>
> **Q1. Why does the Lanczos/Krylov operator help PP-GNNs?** It gives PP-GNNs a better basis for representing each channel’s multi-hop diffusion features under the same hop budget.
>
> **(1) Better conditioning.** For a channel $x_c$, the standard hop bank $[x_c,\ \tilde{L}x_c,\ \ldots,\ \tilde{L}^{m-1}x_c]$ is a monomial basis whose columns are often highly correlated, so the Gram matrix can be ill-conditioned. Lanczos instead constructs an orthonormal basis for the same channel-wise Krylov subspace, and the resulting Ritz atoms are mutually orthogonal. This makes coefficient learning less coupled, reduces cancellation among correlated hop features, and is more stable at small hop budgets.
>
> **(2) Channel-adaptive start vectors preserve the intended filter family.** We use the normalized channel feature $q_{c,1}=x_c/\|x_c\|$ as the Lanczos starting vector, rather than a shared random start. This makes the Krylov subspace exactly $\mathcal{K}_m(\tilde{L},x_c)$, so no degree-$(m-1)$ polynomial response of that channel is discarded. A shared random start instead yields a feature-independent subspace $\mathcal{K}_m(\tilde{L},r)$, which in general preserves only $P_r x_c$, where $P_r$ is the orthogonal projector onto that subspace. Thus, our construction is better aligned with the signal of interest and avoids irreversible information loss.
>
> **(3) More parsimonious coordinates for PP-GNN aggregation.** In the Ritz bank, $Q_c g(T_c) Q_c^\top x_c = \sum_{i=1}^m g(\lambda_{c,i}) z_{c,i}$, where $z_{c,i} := Q_c (u_{c,i}u_{c,i}^\top) Q_c^\top x_c$. Each scalar $g(\lambda_{c,i})$ therefore directly weights one orthogonal spectral atom. When the desired response is concentrated on a few spectral components, the coefficient vector $(g(\lambda_{c,1}), \ldots, g(\lambda_{c,m}))$ is sparse or approximately sparse. By contrast, because each $z_{c,i}$ is generally a dense polynomial combination of $\tilde{L}^k x_c$ for $k=0,\ldots,m-1$, the same response is spread across many correlated monomial hop features. The result is a more parsimonious coordinate system, which is easier for PP-GNN heads to learn under a fixed budget.
>
> **Q2. Can we provide a direct time comparison?** We agree that a direct efficiency illustration is important, and we will add a runtime--accuracy figure on Pokec (https://anonymous.4open.science/r/ICML_30615-C698/README.md) comparing representative MP-GNNs, vanilla PP-GNNs, and our enhanced PP-GNN variants. This figure will make the trade-off explicit: our method does increase wall-clock time relative to the vanilla PP-GNNs, but it does so while substantially improving accuracy, pushing PP-GNNs to a stronger Pareto frontier, rather than negating the scalability benefit of pre-propagation. We will add this figure and the corresponding discussion in the revision.
>
> **Q3. What is original here, and are these operators specific to PP-GNNs?** We agree that the draft should more clearly separate adopted components from original ones. The Jacobi polynomial family itself is not our contribution; we adopt it as a better-conditioned polynomial basis than the standard monomial diffusion basis. Our contribution is its PP-GNN-specific instantiation: we introduce a one-time graph-spectral calibration to set $(\alpha,\beta)$ during preprocessing, rather than treating them as training-time hyperparameters. This distinction matters in PP-GNNs because propagation is performed once and then reused during dense training. Repeatedly tuning $(\alpha,\beta)$ against validation performance would couple preprocessing back to the training loop, increase data-preparation overhead, and undermine the main purpose of PP-GNNs as a decoupled, reusable diffusion pipeline. Thus, our novelty is not a new Jacobi family beyond prior spectral GNN work, but a training-free calibration strategy and PP-GNN-oriented deployment of that family, together with the feature-adaptive Krylov operator and HRP framework.
>
> More broadly, the proposed diffusion operators are portable in principle, but their cost profile depends strongly on the training regime. For PP-GNNs, both calibrated Jacobi diffusion and our feature-adaptive Krylov diffusion are one-time preprocessing costs. For standard MPNNs, calibrated Jacobi is still feasible, but feature-adaptive Krylov becomes much heavier because the Lanczos procedure uses node feature channels as starting vectors; as hidden states change across layers and epochs, the Krylov bases would generally need to be recomputed repeatedly. This is a main reason we focus on PP-GNNs: they are the regime where stronger diffusion operators can be introduced without sacrificing the decoupled preprocessing advantage. Comparing these operators inside MPNNs against spectral GNNs is an interesting future direction, but it is not the setting targeted by this paper.

---

> > ### Author Rebuttal · Reviewer_Xn5G · 2026-04-02
> >
> > Thanks for your detailed response. I will keep my original positive score (5), and hope the discussions on efficiency and originality can be further incorperated into the revised paper.

---

> > > ### Author Response · Authors · 2026-04-02
> > >
> > > Thank you very much for the positive update and for maintaining the encouraging score. We appreciate the reviewer’s helpful suggestions, and in the revised version, we will incorporate the clarifications on efficiency and originality to make these points more explicit and the paper clearer overall.

---

### Official Review · Reviewer_1HkR · 2026-03-13

**Soundness:** 3
**Presentation:** 2
**Significance:** 2
**Originality:** 3
**Overall Recommendation:** 4
**Confidence:** 4

**Summary:**

This paper tackles the expressivity limitations of Pre-Propagation GNNs (PP-GNNs), particularly their performance gap compared to message-passing GNNs on heterophilic graphs. To enhance expressive power without sacrificing training scalability, the authors propose three key architectural improvements. Extensive evaluations across 12 datasets demonstrate that these enhancements allow PP-GNNs to match or exceed the accuracy of state-of-the-art MP-GNNs while preserving pre-propagation efficiency.

**Compliance With Llm Reviewing Policy:**

Affirmed.

**Final Justification:**

Overall, I consider the paper to be sound. In my view, this paper is slightly on the positive side of borderline acceptance. Based on this update, I am increasing my score to 4.

**Key Questions For Authors:**

1. For the Lanczos/Krylov operator on a strongly heterophilic dataset versus a homophilic one, can you visualize the distribution of the learned/selected Ritz values ($\lambda_{c,i}$)? This would empirically prove your hypothesis that your operators adaptively learn higher-frequency components on heterophilic graphs.
2. Were the CMA-ES and Spectral Diversity Score auto-search methods utilized to achieve the benchmark results reported in Section 5? If so, how much does the performance drop if a static blending weight (e.g., $\alpha = 0.5$) and fixed stage epochs are used?

**Limitations:**

Yes.

**Strengths And Weaknesses:**

Strength:
1. Identifying the accuracy-scalability trade-off in PP-GNNs, especially the degradation on heterophilic graphs, is relevant to the graph learning community.
2. The use of Jacobi polynomials and Lanczos/Krylov subspace methods provides a rigorous algebraic solution to the collinearity problem of standard multi-hop diffusion.

Weakness：
1. The paper introduces two distinct robust operators. However, in the main results and the ablation study, they are collectively denoted simply as "+ robust op." It is entirely unclear which operator was used to achieve these results, or if they were somehow combined. This severely diminishes the reproducibility and interpretability of the paper.
2. For massive graphs (e.g., OGB scale), this spatial  memory complexity could be a significant bottleneck, which the authors fail to profile or discuss.
3. While the Krylov/Jacobi operators mathematically allow for high-pass/band-pass filtering, the paper lacks an analysis of what the model actually learns. Do the adaptive Ritz values (Eq. 15) indeed capture high-frequency components on heterophilic graphs? Empirical success is shown, but the theoretical link validating the initial motivation is missing.
4. The author mentions that the expressive power of PPGNNs is still unclear, but the article does not seem to provide a further explanation of this expressive power.

---

> ### Author Rebuttal · Authors · 2026-03-31
>
> We thank the reviewer for the constructive feedback and address the main concerns below.
>
> **Q1. What do the Ritz values capture?** We agree that, beyond accuracy gains, it is important to examine whether the model relies more on non-low-pass components on heterophilic graphs. We first clarify the interpretation level. In our method, Krylov preprocessing is spectrally interpretable: for each channel, Lanczos produces Ritz atoms with associated Ritz values, so each exposed component has a clear spectral meaning. However, the downstream PP-GNN backbone is a node-domain aggregator over the precomputed bank, not an explicit spectral filter. Thus, the plots should be read as evidence about which spectrally meaningful inputs are exposed and used, rather than as an exact spectral decomposition of the final predictor.
>
> We compare a strongly heterophilic dataset (Pokec) with a more homophilic one (Amazon-Computer) using four plots at https://anonymous.4open.science/r/Revisiting-Pre-Propagation-GNNs-Robust-Diffusion-Operators-and-Hidden-State-Re-Propagation-834E/README.md. The two weighted Ritz-value maps show the distribution of Ritz atoms after preprocessing, where the vertical axis is the Ritz value λc,i and color indicates normalized basis weight. On Pokec, we observe noticeable mass near the upper spectral range, indicating that the exposed bank contains substantial high-pass components. On Amazon-Computer, the exposed mass is less concentrated on the high-frequency end but in the middle band.
> The two HOGA attention maps then show how the downstream PP-GNN uses these components. On Pokec, the learned attention is more spread out, suggesting that the model does not rely purely on the low-pass part of the bank. On Amazon-Computer, the attention is much more concentrated on early Ritz indices, consistent with a low-pass preference.
> Taken together, these plots suggest that on the heterophilic graph, preprocessing exposes more non-low-pass spectral content, and the downstream PP-GNN uses the bank in a broader, less purely local manner; on the homophilic graph, the model is more concentrated on early/local components. We view these results as qualitative evidence supporting our motivation, while noting that the final PP-GNN remains a node-domain model rather than an explicit spectral filter.
>
> **Q2. Auto-search application and fixed-weight ablation** We apply CMA-ES to the small graph datasets in Tables 2 and 4. Details of the CMA-ES overhead can be found in the response to Reviewer rbyQ.
> To isolate this overhead, we performed a **no-CMA-ES ablation** with fixed blending factors throughout. Relative to the fully tuned version, the average test drop is modest: 0.38 on heterophilic datasets, 0.31 on homophilic datasets, and 0.34 overall. At the same time, HRP without CMA-ES still improves over robust operators alone by **0.51** on average, while tuning recovers another 0.34. We will add this ablation to make clear that the main gain comes from HRP itself, while CMA-ES provides an additional margin.
>
> **Q 3. Robust operator application detail** We agree that the notation “+ robust op.” was too coarse. In our experiments, each result corresponds to a kernel pair: a first-stage preprocessing kernel and an HRP kernel, selected per backbone and dataset from a small candidate pool. In practice, HRP most often uses Jacobi, while the first stage varies across datasets and backbones.
> We will revise the paper to include a detailed appendix table listing the exact kernel pair used for each result, and release full code/configs upon acceptance to ensure reproducibility.
>
> **Q4. Memory and scalability on massive graphs** We agree that memory can become the dominant bottleneck for PP-GNNs on massive graphs. The main issue is input expansion: preprocessing 𝐾 operators across 𝑅 hops materializes
> 𝐾(𝑅+1) feature matrices. Recent systems work (Yue et al., 2025) identifies this as the key scalability challenge and suggests hierarchical data placement as the practical remedy: use GPU memory when possible, host memory when GPU memory is insufficient, and storage-backed training when host memory is also exceeded. Our paper focuses on the algorithmic side—robust operators and HRP—and does not profile these system-level optimizations. We will revise the paper to acknowledge this limitation explicitly and leave integration with CPU-memory / storage-based training as future work.
>
> **Q5. Expressive power of HRP**
> Chen et al. (2020) show that MP-GNNs are limited mainly as functions on rooted graphs: they induce coarser equivalence classes than MP-GNNs and may miss walk-counting-type interactions. HRP mitigates this because it no longer propagates only the raw features once; it re-propagates task-adaptive hidden states, so each new diffusion step operates on nonlinear summaries learned in the previous stage, and can yield finer distinctions between rooted neighborhoods. We will revise the paper to state this high-level rationale clearly.

---

> > ### Author Rebuttal · Reviewer_1HkR · 2026-04-06
> >
> > Some questions have been solved, but the concerns about computational burden remain.

---

> > > ### Author Response · Authors · 2026-04-06
> > >
> > > Thank you for the thoughtful follow-up and for highlighting the remaining concern about computational burden. We agree that this is an important limitation, and we will make the runtime and memory trade-offs more explicit in the revision, especially for large graphs. We appreciate the reviewer’s constructive feedback.

---

### Official Review · Reviewer_6Jdf · 2026-03-14

**Soundness:** 4
**Presentation:** 3
**Significance:** 4
**Originality:** 4
**Overall Recommendation:** 5
**Confidence:** 3

**Summary:**

The paper introduces pre-propagation GNNs as opposed to MP-GNNs where they perform graph diffusion and per-node dense transformations. The paper proposes graph diffusion and few-shot hidden-state re-propagation during training. The paper provides experiments comparing PP-GNNs to MP-GNNs and other methods on various datasets to show their outperformance and robustness.

**Compliance With Llm Reviewing Policy:**

Affirmed.

**Key Questions For Authors:**

- Have you considered using wavelets which is a step ahead of graph diffusion to capture global and local structure of these graphs in a multiscale fashion?

**Limitations:**

Yes

**Strengths And Weaknesses:**

Strengths:

- The paper is very well written and presented and is easy to understand and follow.
- The idea behind using graph diffusion combined with few-shot hidden-state learning seems novel and interesting.
- The paper provides robust and comprehensive experiments comparing against multiple baselines including MP-GNN across various heterophilic datasets to show their method is solid and robust.
- The paper also provides experiments on homophilic datasets to show robustness not only towards heterophilic graphs.

Weaknesses:

- The paper requires a model schematic so it would be easier for the readers to understand.
- Perhaps considering one step ahead of the graph diffusion which is to use graph wavelets by taking differences in the powers of the diffusion could further improve results in the homophillic graphs which could be a plus point for the paper.

---

> ### Author Rebuttal · Authors · 2026-03-31
>
> Thank you very much for the positive assessment and encouragement. We especially appreciate the reviewer’s recognition of the paper’s clarity, the novelty of combining stronger graph diffusion with few-shot hidden-state re-propagation, and the comprehensive evaluation on both heterophilic and homophilic datasets. We are glad that the reviewer found the overall framework technically solid and well supported empirically.
>
> We also agree with the suggestion that the paper would benefit from a model schematic. In the revision, we will add a figure that clearly illustrates the full pipeline, including the preprocessing diffusion stage, the dense PP-GNN backbone, and the few-shot HRP stages. We believe this will make the method even easier to follow.
>
> Regarding graph wavelets: we agree this is an interesting direction, and we appreciate the suggestion. We did not evaluate wavelet-based preprocessing in the current submission mainly because it typically introduces additional preprocessing design choices, such as the wavelet family and predefined scales. In our setting, these scale choices would likely require nontrivial tuning across datasets, which goes somewhat against our goal of keeping the preprocessing stage lightweight and limiting tuning at preprocessing time. For this reason, we focused on Jacobi/Krylov-based operators, which already enrich the spectral basis while remaining compatible with one-time preprocessing. That said, we agree that wavelet-style multiscale constructions, including difference-based views across diffusion powers, could be a meaningful extension of our framework and an interesting direction for future work.

---

> > ### Author Rebuttal · Reviewer_6Jdf · 2026-04-02
> >
> > Thank you for addressing my concerns. All my concerns have been addressed and I will retain my score of 5.

---

> > > ### Author Response · Authors · 2026-04-02
> > >
> > > Thank you for your thoughtful and encouraging feedback. We sincerely appreciate your recognition of the paper’s clarity, technical soundness, and comprehensive evaluation across both heterophilic and homophilic settings. We are especially grateful for your constructive suggestions. In the revision, we will add a model schematic to make the overall pipeline even easier to follow. We also appreciate your suggestion on wavelet-based extensions, which we view as an interesting direction for future work. We further thank you for confirming that our rebuttal addressed your concerns and for your continued support of our work.

---

### Decision · Program_Chairs · 2026-04-30

**Decision:**

Accept (regular)

**Comment:**

This paper aims to upgrade Pre-Propagation GNNs (PP-GNNs), where graph diffusion is applied first, and then per-node dense transformations are learned. Specifically, the work tackles the expressivity limitations of PP-GNNs and their performance gap compared to more standard MPNNs on heterophilic graphs. This is obtained through modest hidden-state re-propagation and an RNN-based hop aggregator. Extensive evaluations demonstrate that these enhancements allow PP-GNNs to match or exceed the accuracy of state-of-the-art MPNNs while preserving pre-propagation efficiency.

All reviewers agree that the paper makes a meaningful contribution to the field. The work is well-motivated, and the proposed approach is clearly presented and supported by a solid empirical evaluation.
During the rebuttal phase, the authors thoroughly addressed most of the concerns raised by the reviewers. Most of them have updated their assessments and voted in favor of acceptance. In light of this consensus and the demonstrated advantages of the work, I recommend acceptance.